# Application of AHP for the Weighting of Sustainable Development Indicators at the Subnational Level

Abraham Londoño-Pineda [1] , Jose Alejandro Cano [1,*] and Rodrigo Gómez-Montoya [2]

1   Faculty of Economic and Administrative Sciences, Universidad de Medellin, Medellin 050026, Colombia; alondono@udem.edu.co
2   Faculty of Administration, Politécnico Colombiano Jaime Isaza Cadavid, Medellín 050022, Colombia; ragomez@elpoli.edu.co
*   Correspondence: jacano@udem.edu.co

**Abstract:** This article presents an indicator weighting method for constructing composite indices to assess sustainable development at the subnational level. The study uses an analytic hierarchy process (AHP), which is considered relevant, since it establishes links between the indicators that make up the different sustainable development goals (SDG). For this purpose, 28 indicators defined by experts constitute the base to evaluate the progress towards sustainable development of the Aburrá Valley region, located in Antioquia, Colombia. The results show that health, employment, and education indicators obtained higher weights, while environmental indicators received the most reduced weights. Likewise, the model proves to be consistent using a consistency ratio, which generates the possibility of replicating this model at different subnational levels.

**Keywords:** indicator weighting; analytic hierarchy process; inter-thematic frameworks; sustainable development goals; AHP; SDGs

## 1. Introduction

The origin of the sustainable development concept is usually attributed to the "Brundtland Report" published by the World Commission on Environment and Development (WCED) (Holden et al. 2014; Jónsson et al. 2016). This report provided the most popular definition of sustainable development, defining that this is the one who guarantees " . . . the needs of the present without compromising the ability of future generations to meet their own needs" (WCED 1987). Then, Sachs (1980, 1981) stated that sustainable development was not only an economic issue, but that it should be understood as a multidimensional phenomenon, considering social and environmental dimensions (Estenssoro and Devés 2013). Since the Rio Earth Summit 1992, the most common way to define sustainable development has been through the triple bottom line perspective (economic, social, and environmental dimensions—see Figure 1a), which must generate interactions between them (Swarnakar et al. 2021; Ali-Toudert and Ji 2017; Tanguay et al. 2010), and have become the most used way for assessing sustainable development (Bolcárová andOlošta 2015; Shaker and Sirodoev 2016).

In addition to these three dimensions, authors, such as Sepúlveda et al. (2005); Sepúlveda (2008); Toumi et al. (2017) added the institutional dimension to the assessment of sustainable development (see Figure 1b). On the other hand, Holden et al. (2014) consider that the ecological footprint, the human development index, the Gini index, and renewable energy sources as a percentage of total energy sources represent the four variables that allow sustainable development assessment. Recent works, such as those by Londoño and Cruz (2019), use the sustainable development goals (SDGs) as a guide to assess sustainable development because the SDG agenda will be in effect until 2030, and from there, a baseline of 17 SDGs, 169 goals, and 232 indicators will serve as a guide for assessing sustainable development.

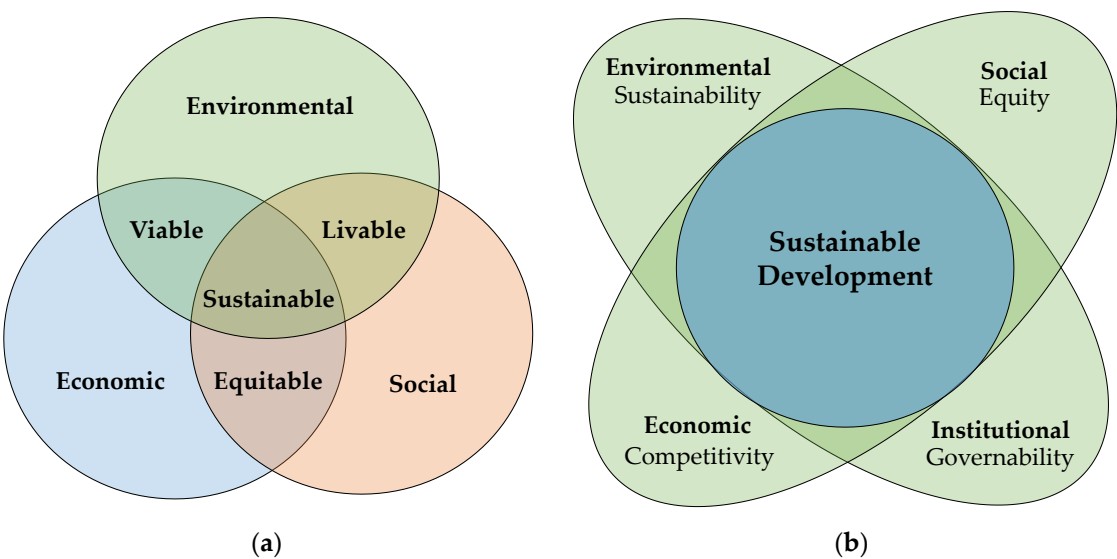

**Figure 1.** Three dimensions of sustainable development (**a**); four dimensions of sustainable development (**b**).

Consequently, the most conventional method for assessing sustainable development has been through the construction of aggregate or composite indices (Boggia and Cortina 2010; Hezri and Hasan 2004; Tanguay et al. 2010) since it supports the evaluation of complex phenomena and facilitates decision-making processes when formulating public policies (Becker et al. 2017; Singh et al. 2012; Ness et al. 2007). Likewise, the aggregate or composite indexes start from a conceptual model and summarize a multidimensional concept in a one-dimensional index (Londoño and Cano 2021; Mauro et al. 2021; Kubule and Blumberga 2020; Rojas et al. 2018).

On the other hand, sustainability studies at different territorial levels are increasing, including subnational regions, cities, municipalities, micro-regions, and even neighborhood groups (Londoño and Cruz 2019; Moreno et al. 2014; Moreno and Fidélis 2015; Schneider et al. 2018; Subramanian et al. 2021), showing the need to assess sustainable development in the different planning and management units (Serna Mendoza et al. 2015). A sign of this is the work by Boggia and Cortina (2010) that assessed sustainable development in 92 municipalities in Umbria, a region in central Italy; Moreno-Pires and Fidélis (2012) presented a similar study in the municipality of Palmela, Portugal; and Lee (2014) did the same for the region of Chiayi, Taiwan. In turn, Serna Mendoza et al. (2015) evaluated sustainable livelihoods in Comuna 1 in Medellin-Colombia using a methodology of aggregate indexes. Something similar was proposed by Phillis et al. (2017), studying the sustainability of 106 cities around the world; Helsinki is the first city in the world in terms of sustainable development, while the capital of the Aburrá Valley (Medellín) is the number 90, and Karachi the last worldwide. Recently, through sustainable indexes, Londoño and Cruz (2019) evaluated the sustainable development of the nine regions that make up the department of Antioquia, Colombia. Likewise, Subramanian et al. (2021) assessed neighborhood sustainability based on the five capital models using SDGs and geographic information systems, quantifying 26 indicators through a bottom-up model applied to the Sha Tin neighborhood in Hong Kong. Consequently, these investigations require indicator weighting processes, which highlights the contribution of this study since it contributes to improving the assessing methodologies for sustainable development at a subnational level.

Therefore, this paper presents an indicator weighting method to construct aggregated indices for evaluating sustainable development at the subnational level, using an AHP and a pairwise comparison technique. The remainder of this paper is organized as follows. Section 2 introduces the proposed method for indicator weighting and the subnational level information used to prove the AHP approach. Section 3 presents the results and

discussions. The paper concludes with a summary and an outlook on future research topics in Section 4.

## 2. Theoretical Background

Assessing sustainable development using aggregated indices requires, in general, the following stages: a conceptual framework development, indicator selection, imputation of missing data, data standardization, indicator weighting, and aggregation, and the sensitivity analysis (Alam et al. 2016; Angeon and Bates 2015; Subramanian et al. 2021; Ciommi et al. 2017; Dobbie and Dail 2013).

The conceptual framework development establishes the approach to analyze the sustainable development assessment, establishing the backgrounds, contexts, and relationships between the economic, social, environmental, and institutional dimensions. The indicator selection for sustainable development assessment should provide an estimate of the achievement of the SDGs, quantify both the determinants and the final impacts of sustainability, and influence the decision-making process with the information they provide. The indicator selection can be supported by several methods and should adopt the following principles: systematic, consistency, independency, measurability, comparability (Wang et al. 2009; Rigo et al. 2020; Shao et al. 2020). Imputation of missing data recognizes the existence of variables that cannot be observed or measured, so an imperfect or approximate measurement can be achieved based on the valid values of other variables or based on a sample. It tends to be recurrent in sustainable development assessment at the subnational level due to the lack of structured and formal sources of information related to the SDGs. Data standardization normalizes the data obtained from the selected indicators so they can be used for quantitative calculations (Shao et al. 2020). Indicator weighting allows a trade-off between multiple indicators and a balance between different sustainable development perspectives, so it is necessary to select the weighting method to define or quantify the importance/relevance of the selected indicators (Rigo et al. 2020; Németh et al. 2019). The aggregation implies aggregating the normalized information of the selected indicators considering their weighting for calculating a total sustainable development score. Sensitivity analysis represents a validation method mainly performed by varying the indicator weights and indicators values to establish how stable the results are when changing the conditions considered in the sustainable development assessment (Shao et al. 2020; Rigo et al. 2020; Allen et al. 2020).

For indicator weighting, weighting methods are classified into equal weighting and rank-order weighting (Shao et al. 2020; Si and Marjanovic-Halburd 2018). In equal weighting, indicator weights are equally assigned, which does not require stakeholder preferences; however, it ignores the relative importance of the criteria (Shao et al. 2020; Wang et al. 2009). In rank-order weighting methods, indicators weights are distributed and influenced by stakeholder perspectives (Si and Marjanovic-Halburd 2018). The rank order weighting methods include subjective weighting method, objective weighting method and combination weighting method (Wang et al. 2009). Subjective weighting methods depend only on the preference of decision-makers, while objective weighting methods emphasize the statistical evaluation of the given data, complex calculation process, and intensive data requirement (Shao et al. 2020). The combination weighting method is used to balance merits and limitations of objective and subjective weighting methods, but the process is complex and not widely-used (Si and Marjanovic-Halburd 2018).

Subjective weighting methods include analytic hierarchy process (AHP), analytic network process (ANP), Delphi method, pairwise comparison, ranking method, SIMOS method, simple multi-attribute rating technique (SMART), step-wise weight assessment ratio analysis (SWARA), swing weighting (SW) (Vavrek 2019; Wang et al. 2009). Objective weighting methods include criteria importance through inter-criteria correlation (CRITIC), entropy method, mean weight (MW), standard deviation (SD), coefficient of variance (CV), statistical variance procedure (SVP), integrated determination of objective criteria weights

(IDOCRIW), criterion impact loss (CILOS), principal component analysis (PCA) (Vavrek 2019; Singh et al. 2009).

For sustainable development assessment, several techniques, such as equal weighting, principal component analysis (PCA), linear regressions, perception surveys, expert criteria, and analytical hierarchy process (AHP) are used and validated (OECD 2008), showing the equal weighting and the principal component analysis (PCA) method a predominance in the literature (Gan et al. 2017; Singh et al. 2012). However, multi-criteria methods are the most appropriated for assessing sustainable development since these problems are multidimensional, involving people, institutions, natural resources, and the environment (Diaz-Balteiro et al. 2017; Munier 2005).

It is emphasized that inter-thematic frameworks, understood as integrated conceptual frameworks where the indicators belonging to a theme are linked to indicators of other sustainable development issues, require indicator-weighting techniques, focusing on the interconnections or relationships existing between the economic, social, environmental, and institutional dimensions (Chebaeva et al. 2021; Iddrisu and Bhattacharyya 2015; Panda et al. 2016). In this regard, the AHP is presented as a valid alternative for the indicator weighting with a focus on inter-thematic frameworks since the problems of sustainable development are multidimensional (Janeiro and Patel 2015; Diaz-Balteiro et al. 2017; Solangi et al. 2021). Likewise, the use of AHP for the indicator weighting in inter-thematic frameworks is justified because it helps to analyze transversely the dimensions forming sustainable development (Gan et al. 2017). Moreover, the AHP method is one of the most popular multiple-criteria decision-making approaches and is used to prioritize or determine the weights of several factors and sub-factors, considering multiple criteria and multiple stakeholder groups for problems of complex scenarios, and capturing subjective and objective evaluation measures that are easy to use and are scalable (Ghorbanzadeh et al. 2019; Nam et al. 2019; Zhang et al. 2019).

Moreover, the AHP is simple to understand and apply to complex issues using uses a nine-point scale to compare criteria relative importance. The AHP decomposes a large problem into smaller sub-problems at hierarchical levels, introduces the comparative importance of the criteria, showing a more reliable representation of the decision goal. The AHP is applicable for quantitative and qualitative criteria and checks the consistency of the decision, thus reducing the bias in the decision-making progression (Shao et al. 2020; Si and Marjanovic-Halburd 2018). Nonetheless, the AHP is flexible and allows adapting to the conditions of each region according to their most pressing needs and problems. In this regard, this study is also convenient for the consolidation of research lines on sustainable development at the local scale, which is evident in the works of Moreno et al. (2014); Moreno and Fidélis (2015).

## 3. Methodology

This study focuses on the Aburrá Valley region belonging to Antioquia state (Colombia) to implement an indicator weighting method for assessing sustainable development at the subnational level. Antioquia is a state located northwest of Colombia and has a territorial extension of 63,612 km$^2$. It also occupies sixth place in extension in Colombia, but it is considered the most populated state with about 6,300,000 inhabitants. The Aburrá Valley is a region representing the administrative-political entity in the state of Antioquia, Colombia. Its core city is Medellín and the other cities belonging to Aburrá Valley are Caldas, La Estrella, Sabaneta, Itagüí, Envigado, Bello, Copacabana, Girardota, and Barbosa (see Figure 2).

The Aburrá Valley has an information center called Metropolitan Information Observatory (MIO) that is in charge of presenting the analysis, processing, and interpretation of indicators that support the institutional management and regional planning, focusing on subjects such as planning and territory, environment, mobility, quality of life, and institutions (AMVA 2019). Therefore, these conditions facilitate the design and implementation of sustainable development assessment methods at the subnational level. Figure 3 shows a

flowchart for the proposed indicator weighing at the subnational level, highlighting the main steps that must be followed, from the indicator selection to the indicator aggregation. Each step is explained in the remainder of this section.

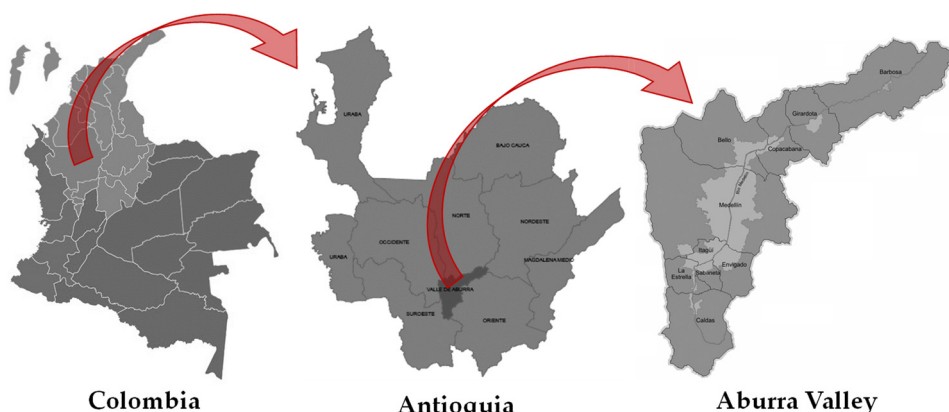

**Figure 2.** Aburrá Valley region.

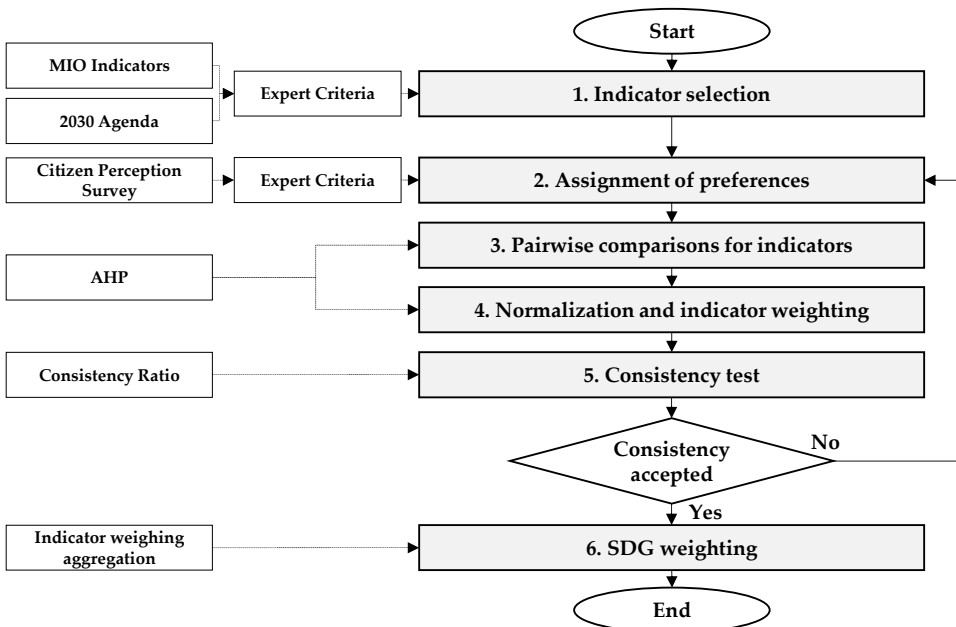

**Figure 3.** Indicator weighing model.

### 3.1. Data and Indicator Description

For the indicator selection stage, indicators must offer an estimation of a specific region's ability to implement the SDGs for sustainable development assessment (State et al. 2019) and must quantify both the determinants and the final impacts of sustainability (Hui et al. 2019). Likewise, indicators must be meaningfully used from the perspective of sustainable development (Da Silva et al. 2020) and must measure progress and support decision-making by providing a simplified view of complex phenomena (Karnauskaite et al. 2019).

As the 2030 Agenda proposes 232 indicators to assess sustainable development, it is challenging to acquire information at the country level and much more difficult at the region level. Therefore, the selection of indicators is based on the information from the MIO and the guidelines of the 2030 Agenda. The goals and indicators that could be used to assess sustainable development were taken from the 2030 Agenda, and the available indicators that can measure the SDGs are listed from the MIO. Then, through expert criteria, indicators from the MIO that effectively measure each of the SDGs established in the 2030

Agenda were selected. As a result of the expert criteria, Table 1 shows the 28 selected indicators to assess sustainable development in the Aburrá Valley, and the description of each indicator can be found in Table A1. Note that indicators for SDG 9, SDG 13, and SDG 14 are not shown in Table 1 due to diverse causes. First, SDG 9 (Industry, innovation, and infrastructure) is not available in the MIO, and the availability of the indicator is essential for the construction of indices (Tanguay et al. 2010; Shaaban and Scheffran 2017). Second, indicators related to SDG 13 (climate action) are not available for all the municipalities of the Aburrá Valley, which is essential to compare them (Londoño 2018) and obtain a unified value for the region. Likewise, in the Aburrá Valley, no indicator covers the topic to be evaluated (Hák et al. 2016). Finally, SDG 14 (life below water) does not apply to the Aburrá Valley because this region is located within the department of Antioquia, so it does not have beaches or access to the sea (Londoño and Cruz 2019).

**Table 1.** Indicators for assessing sustainable development.

| Sustainable Development Goals | Goal Code | Indicator | Indicator Code |
|---|---|---|---|
| 1. No poverty | $SDG_1$ | Percentage of households with at least one unsatisfied basic need (poverty) | $X_1$ |
| | | Percentage of households with two or more unsatisfied basic needs (misery) | $X_2$ |
| 2. Zero hunger | $SDG_2$ | Percentage of households with food insecurity | $X_3$ |
| 3. Good health and well-being | $SDG_3$ | Maternal mortality rate | $X_4$ |
| | | Under-five mortality rate | $X_5$ |
| | | Mortality rate for HIV–AIDS | $X_6$ |
| 4. Quality education | $SDG_4$ | School attendance rate in primary school | $X_7$ |
| | | Secondary school attendance rate | $X_8$ |
| | | Schooling rate in higher education | $X_9$ |
| | | Illiteracy rate from 10 to 14 years old | $X_{10}$ |
| | | Illiteracy rate in people older than 15 years | $X_{11}$ |
| 5. Gender equality | $SDG_5$ | Percentage of women in municipal councils | $X_{12}$ |
| 6 Clean water and sanitation | $SDG_6$ | Aqueduct coverage | $X_{13}$ |
| | | Sewer coverage | $X_{14}$ |
| 7. Affordable and clean energy | $SDG_7$ | Energy coverage | $X_{15}$ |
| | | Gas connection coverage | $X_{16}$ |
| 8. Decent work and economic growth | $SDG_8$ | Unemployment rate | $X_{17}$ |
| | | Underemployment rate | $X_{18}$ |
| | | Dependency ratio | $X_{19}$ |
| 10. Reduced inequalities | $SDG_{10}$ | Indebtedness index | $X_{20}$ |
| | | Internet coverage | $X_{21}$ |
| 11. Sustainable cities and communities | $SDG_{11}$ | Percentage of rural land | $X_{22}$ |
| | | Concentration of particulate material PM2,5 | $X_{23}$ |
| 12. Responsible consumption and production | $SDG_{12}$ | Percentage of solid waste used | $X_{24}$ |
| 15. Life on land | $SDG_{15}$ | Percentage of soil protection | $X_{25}$ |
| 16. Peace, justice and strong institutions | $SDG_{16}$ | Homicide rate | $X_{26}$ |
| | | Kidnapping rate | $X_{27}$ |
| 17. Partnerships for the goals | $SDG_{17}$ | Internet penetration rate | $X_{28}$ |

In the preference assignment stage, it is necessary to assign preferences to sustainable development indicators using expert criteria. The preference assignment process is supported by the citizen perception survey (CPS) of Ipsos Napoleón Franco firm (Medellín Cómo Vamos 2017). The sample size of the CPS was 1502 surveys, and the data collection was based on personal interviews in households. The most relevant life-quality issues for citizens were prioritized considering CPS results. Table 2 shows the relationship between CPS topics, indicators, and SDGs and states that the most relevant issues are health, employment, education, housing, security, feeding, and others (gender equality, environment, and incomes). The indicator relationship with SDGs can be found in Table A2.

**Table 2.** Scale of topics according to the CPS.

| Topic Preferences from CPS | Indicators | SDGs to Which the Indicators Belong |
|---|---|---|
| 1. Health | $X_4, X_5, X_6$ $X_{23}$ | 3. Good health and well-being 11. Sustainable cities and communities |
| 2. Employment | $X_{17}, X_{18}$ | 8. Decent work and economic growth |
| 3. Education | $X_7, X_8, X_9, X_{10}, X_{11}$ | 4. Quality education |
| 4. Housing | $X_1, X_2$ $X_{13}, X_{14}$ $X_{15}, X_{16}$ $X_{28}$ | 1. No poverty 6. Clean water and sanitation 7. Affordable and clean energy 17. Partnerships for the goals |
| 5. Security | $X_{26}, X_{27}$ | 16. Peace, justice, and strong institutions |
| 6. Feeding | $X_3$ | 2. Zero hunger |
| 7. Others: gender equality, environment, incomes | $X_{12}$ $X_{19}, X_{20}, X_{21}$ $X_{22}$ $X_{24}$ $X_{25}$ | 5. Gender equality 10. Reduced inequalities 11. Sustainable cities and communities 12. Responsible consumption and production 15. Life on land |

This information serves as input for the expert criteria to define paired comparisons between indicators. The expert-criteria states that the variables of those indicators associated with social dimensions such as poverty and hunger prevail over environmental dimensions since the Aburrá Valley belongs to a developing country, where the priority is often given to this type of dimension. This premise is supported by the studies of Boggia and Cortina (2010); Bečić et al. (2012); Abou-Ali and Abdelfattah (2013); Serna Mendoza et al. (2015), who indicate that indicators available in developing countries are mainly focused on socioeconomic dimensions. However, in recent years, Colombia has presented advances in health and education and reductions in mortality rates, reflecting a high rate of human development (UNDP 2016).

*3.2. AHP for Indicator Weighting*

After prioritizing CPS indicators and obtaining the performance of indicators from the MIO, the expert criteria define the importance of indicators using the original scale of preferences of Saaty (1980), which have been used by several authors (Kamaruzzaman et al. 2018; Kurka 2013). This scale of preferences presents nine qualifications (see Table 3) and indicates which indicator is more important than another, assigning weights to each one through the AHP.

It is significant to mention that the prioritization of the topics performed by the expert group must provide logical consistency, which implies complying with the preference transitivity criteria. Thus, if the health topic remains a priority to employment, and the employment topic is preferred to education, then the health topic must be preferred to education. Then, the same analysis must be performed with the other topics and checked later by a consistency test.

**Table 3.** Scale for paired comparisons.

| Intensity | Definition | Explanation |
|-----------|------------|-------------|
| 1 | Equal importance between both elements | Two activities contribute equally to the objective |
| 3 | Moderate importance of one over another | Experience and judgment slightly favor one activity over the other |
| 5 | Strong importance | Experience and judgment strongly favor one activity over the other |
| 7 | Very strong importance | Experience and judgment very strongly favor one activity over the other |
| 9 | Absolute importance | Experience and judgment absolutely favor one activity over the other |
| 2, 4, 6, 8 | Intermediate values between adjacent scales | Used to represent the compromise between the priorities listed above |

The AHP was introduced by Saaty (1980) to establish preferences following a format of pairwise comparisons based on a fundamental verbal scale, which aims to define the importance of each variable concerning the others. When paired comparisons are made for $N$ variables, the judgment matrix ($A$) is formed, in which each entry $a_{ij}$ in the matrix is created by comparing the elements of row $A_i$ with the corresponding elements of column $Aj$ so that $A = (a_{ij})$, where $i = 1, 2, \ldots, N$ and $j = 1, 2, \ldots, N$ represent the number of criteria. The question asked is this: is the item in the row more relevant than the item in the column? When comparing the same variable, the rating will be 1, so the main diagonal of $A$ is equal to 1 ($a_{ii} = 1$ or $a_{jj} = 1$). Below the main diagonal appears the inverse ratings to those that appear above it, and in this case, the judgment matrix contains values from 1 to 9 and their corresponding inverse values.

The judgment matrix for indicators must be normalized to obtain the weights for each indicator. For this, following Equation (1), each matrix value is divided over the column total where it belongs. Then, the rows of all the standardized values ($sa_{ij}$) are summed and divided by the number of indicators (in this case, 28) as shown in Equation (2), obtaining the weighting for each indicator ($W_i$). Once the judgment matrix $A$ is completed, the problem becomes a problem of vectors and eigenvalues: $Aw = \lambda w$, where $A$ is the reciprocal matrix of pairwise comparisons, $w$ is the eigenvector that represents the ranking or priority order, and $\lambda$ is the maximum eigenvalue representing a consistency measure of the judgments.

$$sa_{ij} = \frac{a_{ij}}{\sum_{i=1}^{N} a_{ij}} \quad \forall j \in N \tag{1}$$

$$W_i = \frac{\sum_{j=1}^{N} sa_{ij}}{N} \quad \forall i \in N \tag{2}$$

It is necessary to calculate the consistency ratio ($CR$) proposed by Saaty (1990) to verify the consistency of the AHP. As shown in Equation (3), this ratio is based on a consistency index ($CI$) and a random index ($RI$). If $CR < 0.10$, the consistency is reasonable, whereas if the $RC > 0.10$, the hierarchical analysis model is inconsistent, and the preference assignment and pairwise comparisons must be performed again to obtain a reliable indicator weighing model.

$$CR = \frac{CI}{RI} \tag{3}$$

Equation (4) is used for the consistency index, where $\lambda max$ is the maximum eigenvalue obtained by multiplying the total sums vector from the paired comparisons matrix by the weighted values vector from the normalized matrix. On the other hand, $n$ represents the number of elements of the matrix ($N \times N$).

$$CI = \frac{\lambda max - n}{n - 1} \tag{4}$$

The random index depends on the number of elements compared. In this regard, Saaty and Kearns (1985) present a random index for a matrix of up to 10 elements. However, Equation (5) is used for obtaining the random index, where the coefficient 1.98 is used when *n* > 10 (Aguarón and Moreno-Jiménez 2013).

$$RI = \frac{1.98\,(n-2)}{n} \tag{5}$$

Once the weights of the indicators are obtained for assessing sustainable development, the weighting of each SDG is calculated by adding the weights assigned to the indicators of each SDG, so both the indicator weights sum and the SGD weights sum are equal to 1.

## 4. Results and Discussion

This section presents the weighting results for the indicators selected in Section 3.1, which were prioritized with the expert criteria and supported by information from the MIO and the CPS. As a result of the AHP method, Table 4 shows the matrix (*A*) of paired comparisons, which contains values from a 9-point scale to compare indicators' relative importance that corresponds to the original scale of preferences of Saatty.

These values reflect the importance of indicators defined by expert criteria by comparing elements of a row with elements of a column. For example, when comparing $X_4$ with $X_{20}$ (value of row 4 and column 20), a value of 9 was assigned, which means that the indicator $X_4$ (maternal mortality rate) is of absolute importance concerning the indicator $X_{19}$ (Indebtedness index), then absolutely favoring the good health and well-being of pregnant women concerning the level of indebtedness of the municipalities, that is, prioritizing the social dimension over the economic and institutional dimension. Respectively, the value of the element in row 20 and column 4 corresponds to the inverse value of 9 (0.11).

Likewise, when comparing $X_4$ with $X_{23}$ (row 4 and column 23), a value of 1 was assigned, which means that the indicator $X_4$ (Maternal mortality rate) is equally important as the indicator X23 (Concentration of Particulate Material PM2.5) since air quality directly affects the health of citizens, specifically those who are at risk of respiratory diseases. Therefore, it is considered that both indicators contribute equally to the social dimension. Respectively, the element in row 23 and column 4 corresponds to the inverse of 1, which is also equal to 1.

Equation (1) is applied to the values of the matrix (*A*) of paired comparisons to obtain the normalized matrix, and the indicator weighing calculation is obtained based on that matrix using Equation (2). Consequently, Figure 4 shows the weights of the 28 indicators analyzed and the aggregated weights for the SDGs that make up the sustainable development evaluation model for the Aburrá Valley. For its part, the consistency index was 0.033, and the random index was 1.838, obtaining a consistency ratio of 0.018. As the CR < 0.10, then the proposed indicator weighing model is consistent and reliable. Figure 4 also indicates that indicators associated with health ($X_4$: maternal mortality rate; $X_5$: mortality rate in children under 5 years; $X_6$: mortality rate for HIV–AIDS), employment ($X_{17}$: unemployment rate; $X_{18}$: underemployment rate), and education dimensions ($X_7$: school attendance rate in primary school; $X_8$: secondary school attendance rate; $X_9$: schooling rate in higher education; $X_{10}$: illiteracy rate from 10 to 14 years old; $X_{11}$: illiteracy rate in people older than 15 years) were those obtaining higher weights, while those related to environmental dimensions and those belonging to $SDG_{11}$, $SDG_{12}$, and $SDG_{15}$ ($X_{22}$: percentage of rural land; $X_{24}$: percentage of solid waste used; $X_{25}$: percentage of soil protection) presented the lowest weights. The only exception was the indicator $X_{23}$ (concentration of particulate material - PM 2.5), whose weight was higher because, in recent years, the Aburrá Valley has faced several environmental contingencies affecting human health.

**Table 4.** Paired comparisons matrix.

| | X1 | X2 | X3 | X4 | X5 | X6 | X7 | X8 | X9 | X10 | X11 | X12 | X13 | X14 | X15 | X16 | X17 | X18 | X19 | X20 | X21 | X22 | X23 | X24 | X25 | X26 | X27 | X28 |
|---|---|---|---|---|---|---|---|---|---|---|---|---|---|---|---|---|---|---|---|---|---|---|---|---|---|---|---|---|
| X1 | 1 | 1 | 3 | 0.17 | 0.17 | 0.17 | 0.5 | 0.5 | 0.5 | 0.5 | 0.5 | 3 | 1 | 1 | 1 | 1 | 0.25 | 0.25 | 3 | 3 | 3 | 3 | 0.17 | 5 | 5 | 2 | 2 | 1 |
| X2 | 1 | 1 | 3 | 0.17 | 0.17 | 0.17 | 0.5 | 0.5 | 0.5 | 0.5 | 0.5 | 3 | 1 | 1 | 1 | 1 | 0.25 | 0.25 | 3 | 3 | 3 | 3 | 0.17 | 5 | 5 | 2 | 2 | 1 |
| X3 | 0.33 | 0.33 | 1 | 0.11 | 0.11 | 0.11 | 0.2 | 0.2 | 0.2 | 0.2 | 0.2 | 1 | 0.33 | 0.33 | 0.33 | 0.33 | 0.14 | 0.14 | 1 | 1 | 1 | 1 | 0.11 | 1 | 1 | 1 | 1 | 0.33 |
| X4 | 6 | 6 | 9 | 1 | 1 | 1 | 4 | 4 | 4 | 4 | 4 | 9 | 6 | 6 | 6 | 6 | 2 | 2 | 9 | 9 | 9 | 9 | 1 | 9 | 9 | 8 | 8 | 6 |
| X5 | 6 | 6 | 9 | 1 | 1 | 1 | 4 | 4 | 4 | 4 | 4 | 9 | 6 | 6 | 6 | 6 | 2 | 2 | 9 | 9 | 9 | 9 | 1 | 9 | 9 | 8 | 8 | 6 |
| X6 | 6 | 6 | 9 | 1 | 1 | 1 | 4 | 4 | 4 | 4 | 4 | 9 | 6 | 6 | 6 | 6 | 2 | 2 | 9 | 9 | 9 | 9 | 1 | 9 | 9 | 8 | 8 | 6 |
| X7 | 2 | 2 | 5 | 0.25 | 0.25 | 0.25 | 1 | 1 | 1 | 1 | 1 | 5 | 2 | 2 | 2 | 2 | 0.5 | 0.5 | 5 | 5 | 5 | 5 | 0.25 | 5 | 5 | 4 | 4 | 2 |
| X8 | 2 | 2 | 5 | 0.25 | 0.25 | 0.25 | 1 | 1 | 1 | 1 | 1 | 5 | 2 | 2 | 2 | 2 | 0.5 | 0.5 | 5 | 5 | 5 | 5 | 0.25 | 5 | 5 | 4 | 4 | 2 |
| X9 | 2 | 2 | 5 | 0.25 | 0.25 | 0.25 | 1 | 1 | 1 | 1 | 1 | 5 | 2 | 2 | 2 | 2 | 0.5 | 0.5 | 5 | 5 | 5 | 5 | 0.25 | 5 | 5 | 4 | 4 | 2 |
| X10 | 2 | 2 | 5 | 0.25 | 0.25 | 0.25 | 1 | 1 | 1 | 1 | 1 | 5 | 2 | 2 | 2 | 2 | 0.5 | 0.5 | 5 | 5 | 5 | 5 | 0.25 | 5 | 5 | 4 | 4 | 2 |
| X11 | 2 | 2 | 5 | 0.25 | 0.25 | 0.25 | 1 | 1 | 1 | 1 | 1 | 5 | 2 | 2 | 2 | 2 | 0.5 | 0.5 | 5 | 5 | 5 | 5 | 0.25 | 5 | 5 | 4 | 4 | 2 |
| X12 | 0.33 | 0.33 | 1 | 0.11 | 0.11 | 0.11 | 0.2 | 0.2 | 0.2 | 0.2 | 0.2 | 1 | 0.33 | 0.33 | 0.33 | 0.33 | 0.14 | 0.14 | 1 | 1 | 1 | 1 | 0.11 | 1 | 1 | 1 | 1 | 0.33 |
| X13 | 1 | 1 | 3 | 0.17 | 0.17 | 0.17 | 0.5 | 0.5 | 0.5 | 0.5 | 0.5 | 3 | 1 | 1 | 1 | 1 | 0.25 | 0.25 | 3 | 3 | 3 | 3 | 0.17 | 3 | 3 | 2 | 2 | 1 |
| X14 | 1 | 1 | 3 | 0.17 | 0.17 | 0.17 | 0.5 | 0.5 | 0.5 | 0.5 | 0.5 | 3 | 1 | 1 | 1 | 1 | 0.25 | 0.25 | 3 | 3 | 3 | 3 | 0.17 | 3 | 3 | 2 | 2 | 1 |
| X15 | 1 | 1 | 3 | 0.17 | 0.17 | 0.17 | 0.5 | 0.5 | 0.5 | 0.5 | 0.5 | 3 | 1 | 1 | 1 | 1 | 0.25 | 0.25 | 3 | 3 | 3 | 3 | 0.17 | 3 | 3 | 2 | 2 | 1 |
| X16 | 1 | 1 | 3 | 0.17 | 0.17 | 0.17 | 0.5 | 0.5 | 0.5 | 0.5 | 0.5 | 3 | 1 | 1 | 1 | 1 | 0.25 | 0.25 | 3 | 3 | 3 | 3 | 0.17 | 3 | 3 | 2 | 2 | 1 |
| X17 | 4 | 4 | 7 | 0.5 | 0.5 | 0.5 | 2 | 2 | 2 | 2 | 2 | 7 | 4 | 4 | 4 | 4 | 1 | 1 | 7 | 7 | 7 | 7 | 0.5 | 7 | 7 | 6 | 6 | 4 |
| X18 | 4 | 4 | 7 | 0.5 | 0.5 | 0.5 | 2 | 2 | 2 | 2 | 2 | 7 | 4 | 4 | 4 | 4 | 1 | 1 | 7 | 7 | 7 | 7 | 0.5 | 7 | 7 | 6 | 6 | 4 |
| X19 | 0.33 | 0.33 | 1 | 0.11 | 0.11 | 0.11 | 0.2 | 0.2 | 0.2 | 0.2 | 0.2 | 1 | 0.33 | 0.33 | 0.33 | 0.33 | 0.14 | 0.14 | 1 | 1 | 1 | 1 | 0.11 | 1 | 1 | 1 | 1 | 0.33 |
| X20 | 0.33 | 0.33 | 1 | 0.11 | 0.11 | 0.11 | 0.2 | 0.2 | 0.2 | 0.2 | 0.2 | 1 | 0.33 | 0.33 | 0.33 | 0.33 | 0.14 | 0.14 | 1 | 1 | 1 | 1 | 0.11 | 1 | 1 | 1 | 1 | 0.33 |
| X21 | 0.33 | 0.33 | 1 | 0.11 | 0.11 | 0.11 | 0.2 | 0.2 | 0.2 | 0.2 | 0.2 | 1 | 0.33 | 0.33 | 0.33 | 0.33 | 0.14 | 0.14 | 1 | 1 | 1 | 1 | 0.11 | 1 | 1 | 1 | 1 | 0.33 |
| X22 | 0.33 | 0.33 | 1 | 0.11 | 0.11 | 0.11 | 0.2 | 0.2 | 0.2 | 0.2 | 0.2 | 1 | 0.33 | 0.33 | 0.33 | 0.33 | 0.14 | 0.14 | 1 | 1 | 1 | 1 | 0.11 | 1 | 1 | 1 | 1 | 0.33 |
| X23 | 6 | 6 | 9 | 1 | 1 | 1 | 4 | 4 | 4 | 4 | 4 | 9 | 6 | 6 | 6 | 6 | 2 | 2 | 9 | 9 | 9 | 9 | 1 | 9 | 9 | 8 | 8 | 6 |
| X24 | 0.33 | 0.33 | 1 | 0.11 | 0.11 | 0.11 | 0.2 | 0.2 | 0.2 | 0.2 | 0.2 | 1 | 0.33 | 0.33 | 0.33 | 0.33 | 0.14 | 0.14 | 1 | 1 | 1 | 1 | 0.11 | 1 | 1 | 1 | 1 | 0.33 |
| X25 | 0.33 | 0.33 | 1 | 0.11 | 0.11 | 0.11 | 0.2 | 0.2 | 0.2 | 0.2 | 0.2 | 1 | 0.33 | 0.33 | 0.33 | 0.33 | 0.14 | 0.14 | 1 | 1 | 1 | 1 | 0.11 | 1 | 1 | 1 | 1 | 0.33 |
| X26 | 0.5 | 0.5 | 1 | 0.13 | 0.13 | 0.13 | 0.25 | 0.25 | 0.25 | 0.25 | 0.25 | 1 | 0.5 | 0.5 | 0.5 | 0.5 | 0.17 | 0.17 | 1 | 1 | 1 | 1 | 0.13 | 1 | 1 | 1 | 1 | 0.5 |
| X27 | 0.5 | 0.5 | 1 | 0.13 | 0.13 | 0.13 | 0.25 | 0.25 | 0.25 | 0.25 | 0.25 | 1 | 0.5 | 0.5 | 0.5 | 0.5 | 0.17 | 0.17 | 1 | 1 | 1 | 1 | 0.13 | 1 | 1 | 1 | 1 | 0.5 |
| X28 | 1 | 1 | 3 | 0.17 | 0.17 | 0.17 | 0.5 | 0.5 | 0.5 | 0.5 | 0.5 | 3 | 1 | 1 | 1 | 1 | 0.25 | 0.25 | 3 | 3 | 3 | 3 | 0.17 | 3 | 3 | 2 | 2 | 1 |

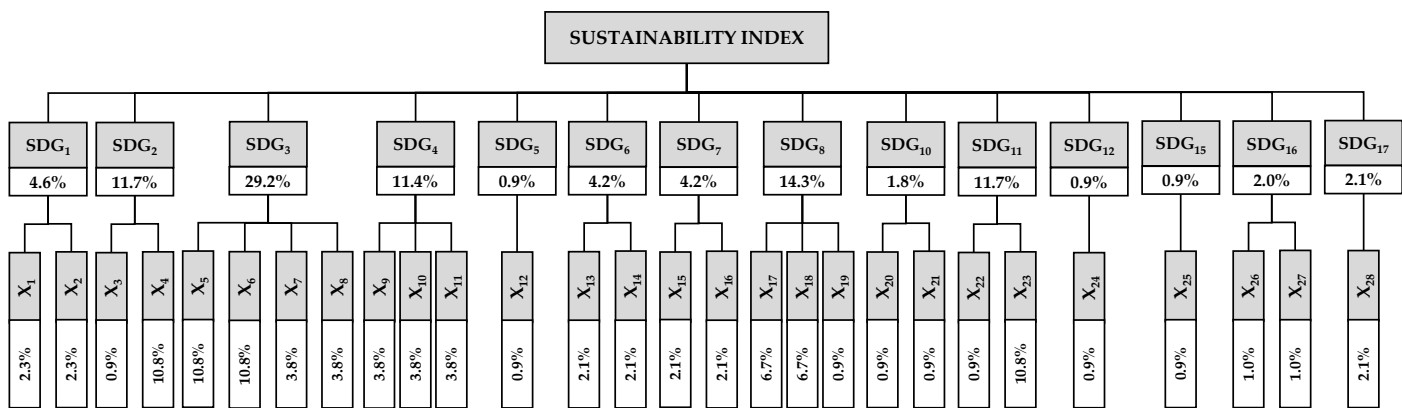

**Figure 4.** Sustainable development index based on indicator weighing.

The results of this work show that the sustainable development priorities for the municipalities of the Aburrá Valley should focus primarily on the health issue, defined by the CPS, since the indicators with the most considerable weight are related to $SDG_3$, as they are the maternal mortality rate, mortality rate for HIV-AIDS, and the under-five mortality rate. The health issue also includes an indicator of $SDG_{11}$ related to concentrations of particulate matter PM2.5 since poor air quality constitutes a public health problem in the Aburrá Valley. This situation is relevant because this region is surrounded by mountains that do not allow adequate air circulation, concentrating particulate material inside the valley.

Moreover, this study prioritizes the issue of employment represented in the indicators that contribute to $SDG_8$. In this sense, the unemployment rate, underemployment rate, and dependency rate worsened due to the confinement generated by the COVID-19 pandemic; therefore, it will continue as a priority in the next quality of life surveys in this region. Similarly, the issue of education occupies the third place of preferences with the indicators related to $SDG_4$, such as the coverage of primary, secondary and higher education, and illiteracy rate indicators. The prioritization of this indicator indicates that governments of the municipalities of the Aburrá Valley must provide proper conditions for citizens to access education, especially higher education, to train citizens for the labor market that continually requires a more skilled workforce.

Accordingly, the indicators with the most considerable weight reflect the priorities of sustainable development in the Aburrá Valley corresponding to $SDG_3$ (health and well-being), $SDG_8$ (decent work and growth economic), and $SDG_4$ (quality education). Therefore, development plans, programs, and the municipal budget of these municipalities should focus on improving health, education, and employment issues.

## 5. Conclusions

This paper introduced an indicator weighting method for assessing sustainable development at the subnational level using an AHP supported by expert criteria. A group of experts assigned preferences based on pairwise comparisons based on the 2030 Agenda and local information from the Metropolitan Information Observatory and the citizen perception survey in the Aburrá Valley. The pairwise comparisons were defined according to the problems of the Aburrá Valley region, so it is coherent that indicators associated with health, employment, and education received the highest weights. On the other hand, indicators related to environmental issues received the lower weights, except for the indicator "particle material concentration (PM 2.5)" due to the environmental contingencies that occurred in this region affecting human health.

Likewise, the proposed indicator weighing model for the Aburrá Valley is consistent, which generates the possibility of replicating this model at different subnational levels. Therefore, the proposed indicator weighing method is flexible and can be adapted to

several features of each region. For this, the adaptation of the model must define the specific problems of each region and assess the progress towards sustainable development under an inter-thematic framework.

This study is novel because it allows prioritizing sustainable development dimensions through a methodology that articulates the technical approach derived from the 17 SDGs of the 2030 agenda (objective) with a participatory approach represented by the CPS (subjective) where citizens prioritize the most relevant life-quality issues. Moreover, a multi-criteria method based on the AHP supports the proposed methodology to lead political decision-making. However, one of the limitations of this paper was not finding available information in the region analyzed to measure the $SDG_9$, $SDG_{13}$, and $SDG_{14}$, so it is suggested that political-administrative entities guide their indicators towards the measurement of the SDGs established in the 2030 Agenda. Future works may implement the proposed methodology in regions, such as the Aburrá Valley, to compare the sustainable development priorities and establish the factors generating variations in the assignment of preferences. Likewise, future studies should consider the uncertainty and vagueness in expert judgment through linguistic evaluations, gray-based methods, and fuzzy logic systems.

**Author Contributions:** Conceptualization, A.L.-P. and J.A.C.; methodology, A.L.-P.; validation, J.A.C.; formal analysis, A.L.-P.; investigation, A.L.-P. and J.A.C.; resources, R.G.-M.; writing—original draft preparation, A.L.-P.; writing—review and editing, J.A.C. and R.G.-M.; visualization, J.A.C. and R.G.-M.; supervision, A.L.-P. and J.A.C. All authors have read and agreed to the published version of the manuscript.

**Funding:** This research received no external funding.

**Institutional Review Board Statement:** Not applicable.

**Informed Consent Statement:** Not applicable.

**Data Availability Statement:** The data presented in this study are available upon request from the corresponding author.

**Conflicts of Interest:** The authors declare no conflict of interest.

## Appendix A

**Table A1.** Description of indicators for assessing sustainable development.

| Indicator Code | Indicator Description |
| --- | --- |
| $X_1$ | It determines whether the essential needs of the population are covered. It is based on the indicators of inadequate housing, housing with critical overcrowding, housing with deficient services, housing with high economic dependence, and housing with school-age children who do not attend school. |
| $X_2$ | It is based on the indicators of inadequate housing, housing with critical overcrowding, housing with deficient services, housing with high economic dependence, and housing with school-age children who do not attend school. In case of not fulfilling two or more of these, it would be in a condition of misery. |
| $X_3$ | It reveals the prevalence of food insecurity at moderate and severe levels. The information is collected through surveys. |
| $X_4$ | It measures the death of a woman during pregnancy, childbirth or during the next 42 days after the end of the pregnancy. It is calculated based on every 100,000 live births. |
| $X_5$ | It represents the number of deaths of children under 5 years for every 1,000 live births for a given year, in each country, territory, or geographic area. |
| $X_6$ | It represents the number of deaths among people with HIV–AIDS per 100,000 inhabitants for a given year, in each country, territory, or geographic area. |
| $X_7$ | Percentage of children between 7 and 11 years old who attend primary school. |
| $X_8$ | Percentage of children between 12 and 17 years old who attend secondary school. |

**Table A1.** *Cont.*

| Indicator Code | Indicator Description |
|---|---|
| $X_9$ | It shows the relationship between students enrolled at the undergraduate level (technical professional, technological, and university) and the projected population between 17 and 21 years old. Therefore, it measures the participation of youth and adults in higher education training programs. |
| $X_{10}$ | It expresses the relative magnitude of the illiterate population and calculates the population between 10 and 14 years old who cannot read and write divided by the population greater than or equal to 15 years. |
| $X_{11}$ | It expresses the relative magnitude of the illiterate population and calculates the population greater than or equal to 15 years old who cannot read and write divided by the population greater than or equal to 15 years. |
| $X_{12}$ | Percentage of women among the total members in the municipal councils. |
| $X_{13}$ | Percentage of households with aqueduct coverage. |
| $X_{14}$ | Percentage of households with sewerage coverage. |
| $X_{15}$ | Percentage of homes with electricity. |
| $X_{16}$ | Percentage of homes with a gas connection. |
| $X_{17}$ | Percentage of workers who are part of the labor force and actively seeking work, but are currently without it. |
| $X_{18}$ | Looks at how well the labor force is being used in terms of skills, experience, and availability to work. People who are classified as underemployed include workers who are highly skilled but working in low-paying or low-skill jobs and part-time workers who would prefer to be full-time. |
| $X_{19}$ | Represents a demographic index and expresses the proportion of people of non-working age, compared with the number of those of working age. |
| $X_{20}$ | It expresses the level of indebtedness of the municipalities. A growing value of this indicator limits the resources available to reduce social inequalities and perform social investment. |
| $X_{21}$ | Percentage of households with internet coverage, which is necessary to reduce technological and social gaps. |
| $X_{22}$ | It expresses the progress of the urbanization of a territory, considering that an increasing value of the indicator threatens the sustainability of cities. |
| $X_{23}$ | It expresses the material with a particle size of fewer than 2.5 microns, known as PM2.5, being the most important in urban pollution since they can penetrate the lungs and pose significant potential risks to health. |
| $X_{24}$ | It expresses the percentage of solid waste that companies separate and dispose of in places specially designed to avoid contamination and risks to human health and the environment. |
| $X_{25}$ | It expresses the percentage of land in the municipality that is protected to avoid its depletion and use in highly polluting activities. |
| $X_{26}$ | It expresses the intentional homicides per 100,000 inhabitants in a territory in a period. |
| $X_{27}$ | It expresses the kidnappings per 100,000 inhabitants in a territory in a period. |
| $X_{28}$ | Percentage of households with at least one member between the ages of 16 and 74 who have internet access or percentage of households with broadband connection. |

**Table A2.** Relationship of indicators with SDGs.

| Indicators | Indicator Relationship with SDGs | SDG |
|---|---|---|
| $X_4$ | This indicator expresses the death of women during pregnancy, childbirth or during the next 42 days after the end of the pregnancy and is directly related to good health and well-being. | |
| $X_5$ | It represents the number of deaths of children under 5 years and is directly related to good health and well-being. | $SDG_3$ |
| $X_6$ | It represents the number of deaths among people with HIV–AIDS and is directly related to good health and well-being. | |
| $X_{23}$ | It expresses the PM2.5, being the most important in urban pollution since they can penetrate the lungs and pose significant potential risks to health. This indicator is directly related to good health and well-being and sustainable cities and communities. | $SDG_{11}$ |

**Table A2.** *Cont.*

| Indicators | Indicator Relationship with SDGs | SDG |
|:---:|:---|:---:|
| $X_{17}$ | It expresses the percentage of workers who are part of the labor force and actively seeking work but are currently without it. It is directly related to decent work and economic growth. | |
| $X_{18}$ | It expresses the labor force who are highly skilled but working in low-paying or low-skill jobs and part-time workers who would prefer to be full-time. It is directly related to decent work and economic growth. | $SDG_8$ |
| $X_{19}$ | It represents a demographic index and expresses the proportion of people of non-working age, compared with the number of those of working age. It is directly related to decent work and economic growth. | |
| $X_7$ | It expresses the percentage of children between 7 and 11 years old who attend primary school. It is directly related to quality education. | |
| $X_8$ | It expresses the Percentage of children between 12 and 17 years old who attend secondary school. It is directly related to quality education. | |
| $X_9$ | It measures the participation of youth and adults in higher education training programs. It is directly related to quality education. | $SDG_4$ |
| $X_{10}$ | It expresses the relative magnitude of the illiterate population. It is directly related to quality education. | |
| $X_{11}$ | It expresses the relative magnitude of the illiterate population. It is directly related to quality education. | |
| $X_1$ | It determines whether the essential needs of the population are covered. It is directly related to poverty. | $SDG_1$ |
| $X_2$ | It is based on the indicators to determine a condition of misery. It is directly related to poverty. | |
| $X_{13}$ | It represents the percentage of households with aqueduct coverage. It is directly related to clean water and sanitation. | |
| $X_{14}$ | It represents the percentage of households with sewerage coverage. It is directly related to clean water and sanitation. | $SDG_6$ |
| $X_{15}$ | It represents the percentage of homes with electricity. It is directly related to affordable and clean energy. | |
| $X_{16}$ | It represents the percentage of homes with a gas connection. It is directly related to affordable and clean energy. | $SDG_7$ |
| $X_{28}$ | It represents the percentage of households with internet connection. One of the targets of SDG 17 is associated with the issue of access to technology and networks. | $SDG_{17}$ |
| $X_{26}$ | It measures the intentional homicides, so it is directly related to peace, justice, and strong institutions. | |
| $X_{27}$ | It measures the kidnappings, so it is directly related to peace, justice, and strong institutions. | $SDG_{16}$ |
| $X_3$ | It reveals the prevalence of food insecurity at moderate and severe levels. It is directly related to zero hunger. | $SDG_2$ |
| $X_{19}$ | It measures the percentage of women among the total members in the Municipal Councils and it is directly related to gender equality. | $SDG_5$ |
| $X_{20}$ | It expresses the level of indebtedness of the municipalities, which limits the resources to reduce social inequalities and perform social investment. It is directly related to reduced inequalities. | |
| $X_{21}$ | It measures the percentage of households with internet coverage to reduce technological and social gaps. It is directly related to reduced inequalities. | $SDG_{10}$ |
| $X_{22}$ | It expresses the progress of the urbanization of a territory, considering that an increasing value of the indicator threatens the sustainability of cities. It is directly related to sustainable cities and communities. | $SDG_{11}$ |
| $X_{24}$ | It measures the proper disposal of solid waste to avoid contamination and risks to human health and the environment. It is directly related to sustainable responsible consumption and production. | $SDG_{12}$ |
| $X_{25}$ | It expresses the percentage of land in the municipality that is protected to avoid its depletion and use in highly polluting activities. It is directly related to life on land. | $SDG_{15}$ |

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
