# Peer review of "Application of AHP for the Weighting of Sustainable Development Indicators at the Subnational Level"

_economies, doi:10.3390/economies9040169_

Round 1

Reviewer 1 Report

The authors of this interesting article present a method for weighting indicators, more precisely for the construction of composite indices useful for assessing sustainable development at the subnational level, the case study discussed was Aburrá Valley region belonging to Antioquia state (Colombia). The topic of this manuscript falls within the objectives of the journal. The topic is well presented, and the English style and grammar are fine, as the text is easy to follow. However, further efforts are needed to make better use of the work. The authors are recommended as follows.
line 130: I would propose to report the three indicators that are not present in table 1.
line 144 and 145: better clarify how the indicators are compared.
line 243 to 251 : I would move the content of this part to an earlier section, even in the introduction.

Author Response

Dear Evay Yuan                                                                                             Economics
MDPI

We communicate with you through this letter in order to outline every change made in the article economies-1394433 “Application of AHP for the Weighting of Sustainable Development Indicators at the Subnational Level” based on the observations and suggestions provided by the reviewers (Reviewer 1, Reviewer 2). All changes and modifications made to the manuscript are highlighted in yellow in the revised version.

Comments from author to reviewers:
-Reviewer 1

The authors of this interesting article present a method for weighting indicators, more precisely for the construction of composite indices useful for assessing sustainable development at the subnational level, the case study discussed was Aburrá Valley region belonging to Antioquia state (Colombia). The topic of this manuscript falls within the objectives of the journal. The topic is well presented, and the English style and grammar are fine, as the text is easy to follow. However, further efforts are needed to make better use of the work. The authors are recommended as follows.

line 130: I would propose to report the three indicators that are not present in table 1.

R// A paragraph is added to explain why indicators related to SDG 9, SDG 13 and SDG 14 were not considered (lines 220-230):

“Note that indicators for SDG 9, SDG 13, and SDG 14 are not shown in Table 1 due to diverse causes. First, SDG 9 (Industry, innovation, and infrastructure) is not available in the MIO, and the availability of the indicator is essential for the construction of indices (Tanguay et al. 2010; Shaaban and Scheffran 2017). Second, indicators related to SDG 13 (Climate Action) are not available for all the municipalities of the Aburrá Valley, which is essential to compare them (Londoño 2018) and obtain a unified value for the region. Likewise, in the Aburrá Valley, no indicator covers the topic to be evaluated (Hák, Janousková, and Moldan 2016). Finally, SDG 14 (Life below water) does not apply to the Aburrá Valley because this region is located within the department of Antioquia, so it does not have beaches or access to the sea (Londoño and Cruz 2019).”

line 144 and 145: better clarify how the indicators are compared.

R// A column is added to Table 2 explaining how these indicators contribute directly to each SDG (line 246).

Topic preferences from CPS

Indicators

Indicator relationship with SDGs

SDGs to which the indicators belong

1. Health

X4

This indicator expresses the death of women during pregnancy, childbirth or during the next 42 days after the end of the pregnancy and is directly related to good health and well-being.

3. Good health and well-being

X5

It represents the number of deaths of children under 5 years and is directly related to good health and well-being.

X6

It represents the number of deaths among people with HIV-AIDS and is directly related to good health and well-being.

X23

It expresses the PM2.5, being the most important in urban pollution since they can penetrate the lungs and pose significant potential risks to health. This indicator is directly related to good health and well-being and sustainable cities and communities.

11. Sustainable cities and communities

2. Employment

X17

It expresses the percentage of workers who are part of the labor force and actively seeking work but are currently without it. It is directly related to decent work and economic growth.

8. Decent work and economic growth

X18

It expresses the labor force who are highly skilled but working in low-paying or low-skill jobs and part-time workers who would prefer to be full-time. It is directly related to decent work and economic growth.

X19

It represents a demographic index and expresses the proportion of people of non-working age, compared with the number of those of working age. It is directly related to decent work and economic growth.

3. Education

X7

It expresses the percentage of children between 7 and 11 years old who attend primary school. It is directly related to quality education.

4. Quality education

X8

It expresses the Percentage of children between 12 and 17 years old who attend secondary school. It is directly related to quality education.

X9

It measures the participation of youth and adults in higher education training programs. It is directly related to quality education.

X10

It expresses the relative magnitude of the illiterate population. It is directly related to quality education.

X11

It expresses the relative magnitude of the illiterate population. It is directly related to quality education.

4. Housing

X1

It determines whether the essential needs of the population are covered. It is directly related to poverty.

1. No poverty

X2

It is based on the indicators to determine a condition of misery. It is directly related to poverty.

X13

It represents the percentage of households with aqueduct coverage. It is directly related to clean water and sanitation.

6 Clean water and sanitation

X14

It represents the percentage of households with sewerage coverage. It is directly related to clean water and sanitation.

X15

It represents the percentage of homes with electricity. It is directly related to affordable and clean energy.

7. Affordable and clean energy

X16

It represents the percentage of homes with a gas connection. It is directly related to affordable and clean energy.

X28

It represents the percentage of households with internet connection. One of the targets of SDG 17 is associated with the issue of access to technology and networks.

17. Partnerships for the goals

5. Security

X26

It measures the intentional homicides, so it is directly related to peace, justice, and strong institutions.

16. Peace, justice, and strong institutions

X27

It measures the kidnappings, so it is directly related to peace, justice, and strong institutions.

6. Feeding

X3

It reveals the prevalence of food insecurity at moderate and severe levels. It is directly related to zero hunger.

2. Zero Hunger

7. Others: gender equality, environment, incomes

X19

It measures the percentage of women among the total members in the Municipal Councils and it is directly related to gender equality.

5. Gender equality

X20

It expresses the level of indebtedness of the municipalities, which limits the resources to reduce social inequalities and perform social investment. It is directly related to reduced inequalities.

10. Reduced inequalities

X21

It measures the percentage of households with internet coverage to reduce technological and social gaps. It is directly related to reduced inequalities.

X22

It expresses the progress of the urbanization of a territory, considering that an increasing value of the indicator threatens the sustainability of cities. It is directly related to sustainable cities and communities.

11. Sustainable cities and communities

X24

It measures the proper disposal of solid waste to avoid contamination and risks to human health and the environment. It is directly related to sustainable responsible consumption and production.

12. Responsible consumption and production

X25

It expresses the percentage of land in the municipality that is protected to avoid its depletion and use in highly polluting activities. It is directly related to life on land.

15. Life on land

line 243 to 251 : I would move the content of this part to an earlier section, even in the introduction.

R// It was decided to remove this paragraph from the results section. In the introduction, several paragraphs were added to deepen on the methods for weighting criteria (lines 84-115).

“For indicator weighting, weighting methods are classified into equal weighting and rank-order weighting (Shao et al. 2020; Si and Marjanovic-Halburd 2018). In equal weighting, indicator weights are equally assigned, which does not require stakeholder preferences; however, it ignores the relative importance of the criteria (Shao et al. 2020; Wang et al. 2009). In rank-order weighting methods, indicators weights are distributed and influenced by stakeholder perspectives (Si and Marjanovic-Halburd 2018). The rank-order weighting methods include subjective weighting method, objective weighting method and combination weighting method (Wang et al. 2009). Subjective weighting methods depend only on the preference of decision-makers, while objective weighting methods emphasize the statistical evaluation of the given data, complex calculation process and intensive data requirement (Shao et al. 2020). Combination weighting method is used to balance merits and limitations of objective and subjective weighting methods, but the process is complex and not widely-used (Si and Marjanovic-Halburd 2018).”

“Subjective weighting methods include Analytic Hierarchy Process (AHP), Analytic Network Process (ANP), Best Worst Method (BWM), Delphi method, DEMATEL, Direct given method, Eigenvector method, Expert judgement, Full consistency method Step-Wise Weight Assessment Ratio Analysis (SWARA), Fuzzy preference programming, Least-square method, Quality function deployment (QFD), Simos method, Simple Multi-Attribute Rating Technique (SMART), Swing method. Objective weighting methods include Data envelopment analysis (DEA), Divergence measure method, Entropy method, Hesitant fuzzy entropy measure, Least mean square (LMS) method, Linear programming technique for multidimensional analysis of preference (LINMAP), Maximizing deviation method, Minmax deviation method, Multiple correlation coefficient, Ordered weight, Preference selection index, Principal component analysis (PCA), Relative preference relation, Time sequence weight, Technique for Order of Preference for Similarity Ideal Solution TOPSIS method, Variation coefficient, Vertical and horizontal method. Combination weighting methods include Additive synthesis, AHP and entropy method, Direct given and ordered weight, Entropy method and divergence measure method, Expert judgement and entropy method, Expert judgement and maximizing deviation method, Expert judgement and maximizing distance method, Expert judgement and statistical variance method, Multiplication synthesis, Optimal weighting based on minimum bias, Optimal weighting based on sum of squares (Wang et al. 2009; L. J. Zhang et al. 2020).”

Comments from author to reviewers:
-Reviewer 2

I am glad I have this opportunity to review your manuscript. Based on its title/abstract it looks very interesting. Anyway, my comments/notes are the follows:

Introduction. Stages of "assessing sustainable development using aggregated indices requires"... could be more described (specify what is the content of each stage by 1-2 sentences).

R// The following paragraph is added to describe each of the steps required for assessing sustainable development (lines 60-83)

“The conceptual framework development establishes the approach to analyze the sustainable development assessment, establishing the backgrounds, contexts, and relationships between the economic, social, environmental, and institutional dimensions. The indicator selection for sustainable development assessment should provide an estimate of the achievement of the SDGs, quantify both the determinants and the final impacts of sustainability, and influence the decision-making process with the information they provide. The indicator selection can be supported by several methods and should adopt the following principles: systematic, consistency, independency, measurability, comparability  (Wang et al. 2009; Rigo et al. 2020; Shao et al. 2020). Imputation of missing data recognizes the existence of variables that cannot be observed or measured, so an imperfect or approximate measurement can be achieved based on the valid values of other variables or based on a sample. It tends to be recurrent in sustainable development assessment at the subnational level due to the lack of structured and formal sources of information related to the SDGs. Data standardization normalizes the data obtained from the selected indicators so they can be used for quantitative calculations (Shao et al. 2020). Indicator weighting allows a trade-off between multiple indicators and a balance between different sustainable development perspectives, so it is necessary to select the weighting method to define or quantify the importance/relevance of the selected indicators (Rigo et al. 2020; Németh et al. 2019). The aggregation implies aggregating the normalized information of the selected indicators considering their weighting for calculating a total sustainable development score. Sensitivity analysis represents a validation method mainly performed by varying the indicator weights and indicators values to establish how stable the results are when changing the conditions considered in the sustainable development assessment (Shao et al. 2020; Rigo et al. 2020; Allen et al. 2020).”

Introduction. • three dimensions could be illustrated by any figure

R// Figure 1 is added to illustrate the three dimensions and four dimensions of sustainable development (line 46)

(a)

(b)

Figure 1. Three dimensions of sustainable development (a); four dimensions of sustainable development (b)

Introduction. • "For indicator weighting, several techniques..." - it is a poorly prepared overview of the methods used for criteria weighting (there are many other methods which could be also used). From my point of view, firstly should be these methods presented in general and after that, it could be more focused on one selected method (AHP),

R// We added two paragraphs dedicated to an overview of the methods used for criteria weighting (lines 84-115)

“For indicator weighting, weighting methods are classified into equal weighting and rank-order weighting (Shao et al. 2020; Si and Marjanovic-Halburd 2018). In equal weighting, indicator weights are equally assigned, which does not require stakeholder preferences; however, it ignores the relative importance of the criteria (Shao et al. 2020; Wang et al. 2009). In rank-order weighting methods, indicators weights are distributed and influenced by stakeholder perspectives (Si and Marjanovic-Halburd 2018). The rank-order weighting methods include subjective weighting method, objective weighting method and combination weighting method (Wang et al. 2009). Subjective weighting methods depend only on the preference of decision-makers, while objective weighting methods emphasize the statistical evaluation of the given data, complex calculation process and intensive data requirement (Shao et al. 2020). Combination weighting method is used to balance merits and limitations of objective and subjective weighting methods, but the process is complex and not widely-used (Si and Marjanovic-Halburd 2018).”

“Subjective weighting methods include Analytic Hierarchy Process (AHP), Analytic Network Process (ANP), Best Worst Method (BWM), Delphi method, DEMATEL, Direct given method, Eigenvector method, Expert judgement, Full consistency method - Step-Wise Weight Assessment Ratio Analysis (SWARA), Fuzzy preference programming, Least-square method, Quality function deployment (QFD), Simos method, Simple Multi-Attribute Rating Technique (SMART), Swing method. Objective weighting methods include Data envelopment analysis (DEA), Divergence measure method, Entropy method, Hesitant fuzzy entropy measure, Least mean square (LMS) method, Linear programming technique for multidimensional analysis of preference (LINMAP), Maximizing deviation method, Minmax deviation method, Multiple correlation coefficient, Ordered weight, Preference selection index, Principal component analysis (PCA), Relative preference relation, Time sequence weight, Technique for Order of Preference for Similarity Ideal Solution TOPSIS method, Variation coefficient, Vertical and horizontal method. Combination weighting methods include Additive synthesis, AHP and entropy method, Direct given and ordered weight, Entropy method and divergence measure method, Expert judgement and entropy method, Expert judgement and maximizing deviation method, Expert judgement and maximizing distance method, Expert judgement and statistical variance method, Multiplication synthesis, Optimal weighting based on minimum bias, Optimal weighting based on sum of squares (Wang et al., 2009; Zhang et al., 2020).”

Likewise, a paragraph is added to describe the functionality and practicality of the AHP for the weighting of indicators (lines 141-147):

“Moreover, the AHP is simple to understand and apply to complex issues using uses a 9-point scale to compare criteria relative importance. The AHP decomposes a large problem into smaller sub-problems at hierarchical levels, introduces the comparative importance of the criteria, showing a more reliable representation of the decision goal. The AHP is applicable for quantitative and qualitative criteria and checks the consistency of the decision, thus reducing the bias in the decision-making progression (Shao et al., 2020; Si & Marjanovic-Halburd, 2018).”

Indicator weighting for sustainable development. •         this is/could be a part of the methodology, not the theoretical section,

R // The name of the section “Indicator weighting for sustainable development” is changed to “Methodology” (line 180)

Indicator weighting for sustainable development. •         Figure 2 - correct "Consistency accepted" where "y" is on a new line,

R // The word “Consistency” is corrected to present it in a single line (line 202)

Indicator weighting for sustainable development. •         Table 1 - I expect more information about each criterion used (some base characteristics could be added, e.g. type of variable, source, some moment characteristics, ...),

R // A column with the description of each indicator is added to Table 1 (line 231)

Sustainable Development Goals

Goal Code

Indicator

Indicator description

Indicator Code

1. No poverty

SDG1

Percentage of households with at least one unsatisfied basic need (Poverty)

It determines whether the essential needs of the population are covered. It is based on the indicators of inadequate housing, housing with critical overcrowding, housing with deficient services, housing with high economic dependence, housing with school-age children who do not attend school.

X1

Percentage of households with two or more unsatisfied basic needs (Misery)

It is based on the indicators of inadequate housing, housing with critical overcrowding, housing with deficient services, housing with high economic dependence, housing with school-age children who do not attend school. In case of not fulfilling two or more of these, it would be in a condition of misery.

X2

2. Zero Hunger

SDG2

Percentage of households with food insecurity

It reveals the prevalence of food insecurity at moderate and severe levels. The information is collected through surveys.

X3

3. Good health and well-being

SDG3

Maternal mortality rate

It measures the death of a woman during pregnancy, childbirth or during the next 42 days after the end of the pregnancy. It is calculated based on every 100.000 live births.

X4

Under-five mortality rate

It represents the number of deaths of children under 5 years for every 1.000 live births for a given year, in each country, territory, or geographic area.

X5

Mortality rate for HIV-AIDS

It represents the number of deaths among people with HIV-AIDS per 100.000 inhabitants for a given year, in each country, territory, or geographic area.

X6

4. Quality education

SDG4

School attendance rate in primary school

Percentage of children between 7 and 11 years old who attend primary school

X7

Secondary school attendance rate

Percentage of children between 12 and 17 years old who attend secondary school

X8

Schooling rate in higher education

It shows the relationship between students enrolled at the undergraduate level (technical professional, technological, and university) and the projected population between 17 and 21 years old. Therefore, it measures the participation of youth and adults in higher education training programs.

X9

Illiteracy rate from 10 to 14 years old

It expresses the relative magnitude of the illiterate population and calculates the population between 10 and 14 years old who cannot read and write divided by the population greater than or equal to 15 years.

X10

Illiteracy rate in people older than 15 years

It expresses the relative magnitude of the illiterate population and calculates the population greater than or equal to 15 years old who cannot read and write divided by the population greater than or equal to 15 years.

X11

5. Gender equality

SDG5

Percentage of women in municipal councils

Percentage of women among the total members in the Municipal Councils.

X12

6 Clean water and sanitation

SDG6

Aqueduct coverage

Percentage of households with aqueduct coverage

X13

Sewer coverage

Percentage of households with sewerage coverage

X14

7. Affordable and clean energy

SDG7

Energy coverage

Percentage of homes with electricity

X15

Gas connection coverage

Percentage of homes with a gas connection

X16

8. Decent work and economic growth

SDG8

Unemployment rate

Percentage of workers who are part of the labor force and actively seeking work but are currently without it

X17

Underemployment rate

Looks at how well the labor force is being used in terms of skills, experience, and availability to work. People who are classified as underemployed include workers who are highly skilled but working in low-paying or low-skill jobs and part-time workers who would prefer to be full-time.

X18

Dependency ratio

Represents a demographic index and expresses the proportion of people of non-working age, compared with the number of those of working age.

X19

10. Reduced inequalities

SDG10

Indebtedness index

It expresses the level of indebtedness of the municipalities. A growing value of this indicator limits the resources available to reduce social inequalities and perform social investment.

X20

Internet coverage

Percentage of households with internet coverage, which is necessary to reduce technological and social gaps.

X21

11. Sustainable cities and communities

SDG11

Percentage of rural land

It expresses the progress of the urbanization of a territory, considering that an increasing value of the indicator threatens the sustainability of cities.

X22

Concentration of Particulate Material PM2,5

It expresses the material with a particle size of fewer than 2.5 microns, known as PM2.5, being the most important in urban pollution since they can penetrate the lungs and pose significant potential risks to health.

X23

12. Responsible consumption and production

SDG12

Percentage of solid waste used

It expresses the percentage of solid waste that companies separate and dispose of in places specially designed to avoid contamination and risks to human health and the environment.

X24

15. Life on land

SDG15

Percentage of soil protection

It expresses the percentage of land in the municipality that is protected to avoid its depletion and use in highly polluting activities.

X25

16. Peace, justice and strong institutions

SDG16

Homicide rate

It expresses the intentional homicides per 100.000 inhabitants in a territory in a period.

X26

Kidnapping rate

It expresses the kidnappings per 100.000 inhabitants in a territory in a period.

X27

17. Partnerships for the goals

SDG17

Internet penetration rate

Percentage of households with at least one member between the ages of 16 and 74 who have internet access or percentage of households with broadband connection.

X28

Pairwise Comparisons •              I recommend changing the title of this section and do not divide next into short sub-sections (Normalized Matrix and Indicator Weighing Calculation, Consistency Test, ...). Is totally useless to have a section including one paragraph (Indicator Aggregation for SDG Weighing). They could be presented as steps of the calculation procedure, but for sure not as sub-sections.

R // The structure of the Methodology section is modified considering only two subsections: 2.1. Data and Indicator Description (line 204) and 2.2 AHP for indicator weighting (line 258)

Pairwise Comparisons •              you present your option (AHP, consistency test) as the only one possible option how to do it, there are many other approaches that are not considered or mentioned at least,

R// We added two paragraphs dedicated to an overview of the methods used for criteria weighting (lines 84-115)

“For indicator weighting, weighting methods are classified into equal weighting and rank-order weighting (Shao et al. 2020; Si and Marjanovic-Halburd 2018). In equal weighting, indicator weights are equally assigned, which does not require stakeholder preferences; however, it ignores the relative importance of the criteria (Shao et al. 2020; Wang et al. 2009). In rank-order weighting methods, indicators weights are distributed and influenced by stakeholder perspectives (Si and Marjanovic-Halburd 2018). The rank-order weighting methods include subjective weighting method, objective weighting method and combination weighting method (Wang et al. 2009). Subjective weighting methods depend only on the preference of decision-makers, while objective weighting methods emphasize the statistical evaluation of the given data, complex calculation process and intensive data requirement (Shao et al. 2020). Combination weighting method is used to balance merits and limitations of objective and subjective weighting methods, but the process is complex and not widely-used (Si and Marjanovic-Halburd 2018).”

“Subjective weighting methods include Analytic Hierarchy Process (AHP), Analytic Network Process (ANP), Best Worst Method (BWM), Delphi method, DEMATEL, Direct given method, Eigenvector method, Expert judgement, Full consistency method - Step-Wise Weight Assessment Ratio Analysis (SWARA), Fuzzy preference programming, Least-square method, Quality function deployment (QFD), Simos method, Simple Multi-Attribute Rating Technique (SMART), Swing method. Objective weighting methods include Data envelopment analysis (DEA), Divergence measure method, Entropy method, Hesitant fuzzy entropy measure, Least mean square (LMS) method, Linear programming technique for multidimensional analysis of preference (LINMAP), Maximizing deviation method, Minmax deviation method, Multiple correlation coefficient, Ordered weight, Preference selection index, Principal component analysis (PCA), Relative preference relation, Time sequence weight, Technique for Order of Preference for Similarity Ideal Solution TOPSIS method, Variation coefficient, Vertical and horizontal method. Combination weighting methods include Additive synthesis, AHP and entropy method, Direct given and ordered weight, Entropy method and divergence measure method, Expert judgement and entropy method, Expert judgement and maximizing deviation method, Expert judgement and maximizing distance method, Expert judgement and statistical variance method, Multiplication synthesis, Optimal weighting based on minimum bias, Optimal weighting based on sum of squares (Wang et al., 2009; Zhang et al., 2020).”

Likewise, a paragraph is added to describe the functionality and practicality of the AHP for the weighting of indicators (lines 141-147):

“Moreover, the AHP is simple to understand and apply to complex issues using uses a 9-point scale to compare criteria relative importance. The AHP decomposes a large problem into smaller sub-problems at hierarchical levels, introduces the comparative importance of the criteria, showing a more reliable representation of the decision goal. The AHP is applicable for quantitative and qualitative criteria and checks the consistency of the decision, thus reducing the bias in the decision-making progression (Shao et al., 2020; Si & Marjanovic-Halburd, 2018).”

Indicator weighting for sustainable development. •         this section should be marked as "Methodology" or "Material and Methods", clearly should be divided into sections focusing on the subject (region), data and criteria description, methods used.

R// The name of the section “Indicator weighting for sustainable development” is changed to “Methodology” (line 180). The structure of the Methodology section is modified considering only two subsections: 2.1. Data and Indicator Description (line 204) and 2.2 AHP for indicator weighting (line 258)

Results and Discussion •             Table 4 - without any description,

R // We improve and deepen the description of Table 4 (lines 314-331):

“As a result of the AHP method, Table 4 shows the matrix (A) of paired comparisons, which contains values from a 9-point scale to compare indicators' relative importance that corresponds to the original scale of preferences of Saatty.”

“These values reflect the importance of indicators defined by expert criteria by comparing elements of a row with elements of a column. For example, when comparing X4 with X20 (value of row 4 and column 20), a value of 9 was assigned, which means that the indicator X4 (Maternal mortality rate) is of absolute importance concerning the indicator X19 (Indebtedness index), then absolutely favoring the good health and well-being of pregnant women concerning the level of indebtedness of the municipalities, that is, prioritizing the social dimension over the economic and institutional dimension. Respectively, the value of the element in row 20 and column 4 corresponds to the inverse value of 9 (0,11).”

“Likewise, when comparing X4 with X23 (row 4 and column 23), a value of 1 was assigned, which means that the indicator X4 (Maternal mortality rate) is equally important as the indicator X23 (Concentration of Particulate Material PM2.5) since air quality directly affects the health of citizens, specifically those who are at risk of respiratory diseases. Therefore, it is considered that both indicators contribute equally to the social dimension. Respectively, the element in row 23 and column 4 corresponds to the inverse of 1, which is also equal to 1.”

Results and Discussion •             Table 5 - totally useless (it is only an intermediate step that is ok, but it is not necessary to describe it by a table),

R// Table 5 is eliminated, and it is mentioned as a complement to the description of Table 4 how the normalized values are calculated (lines 334-336).

“Equation (1) is applied to the values of the matrix (A) of paired comparisons to obtain the Normalized Matrix, and the Indicator Weighing Calculation is obtained based on that matrix using Equation (2).”

Results and Discussion • you are not interpreting the results obtained at all, you just put some numbers there and advocate the use of the AHP method (which should be a part of the previous section),

R// Three paragraphs are added to interpret the results obtained (lines 355-378):

“The results of this work show that the sustainable development priorities for the municipalities of the Aburrá Valley should focus primarily on the health issue, defined by the CPS, since the indicators with the most considerable weight are related to SDG3, as they are the maternal mortality rate, mortality rate for HIV-AIDS, and the under-five mortality rate. The health issue also includes an indicator of SDG 11 related to concentrations of particulate matter PM2.5 since poor air quality constitutes a public health problem in the Aburrá Valley. This situation is relevant because this region is surrounded by mountains that do not allow adequate air circulation, concentrating particulate material inside the valley.”

“Moreover, this study prioritizes the issue of employment represented in the indicators that contribute to SDG8. In this sense, the unemployment rate, underemployment rate, and dependency rate worsened due to the confinement generated by the Covid-19 pandemic; therefore, it will continue as a priority in the next quality of life surveys in this region. Similarly, the issue of education occupies the third place of preferences with the indicators related to SDG4, such as the coverage of primary, secondary and higher education, and illiteracy rate indicators. The prioritization of this indicator indicates that governments of the municipalities of the Aburrá Valley must provide proper conditions for citizens to access education, especially higher education, to train citizens for the labor market that continually requires a more skilled workforce.”

“Accordingly, the indicators with the most considerable weight reflect the priorities of sustainable development in the Aburrá Valley corresponding to SDG3 (Health and well-being), SDG8 (Decent work and growth economic), and SDG4 (Quality education). Therefore, development plans, programs, and the municipal budget of these municipalities should focus on improving health, education, and employment issues.”

Results and Discussion •             "This work was focused on indicator..." - the same problem, this should be a part of other section, real discussion is missing (you do not have any results interpretation to discuss),

R// The entire paragraph was moved and placed at the end of the Introduction section (lines 156-173)

Conclusions •   "However, the novelty of this study is providing an indicator weighting model that allows the assessment of sustainable development at the subnational level." - really? You just choose one method and applied it to one case study, this is novelty?

R// The novelty of this study is highlighted in the conclusions in a new paragraph added (lines 396-401):

“This study is novel because it allows prioritizing sustainable development dimensions through a methodology that articulates the technical approach derived from the 17 SDGs of the 2030 agenda (objective) with a participatory approach represented by the CPS (subjective) where citizens prioritize the most relevant life-quality issues. Moreover, a multi-criteria method based on the AHP supports the proposed methodology to lead political decision-making.”

Conclusions •   "Future research should focus..." - it should be a part of this manuscript, in the current version it looks like you just select a part of your research for publishing it separately.

R// The future research suggestion is modified in the conclusions, showing future lines of research that do not depend directly on the stages of our research (lines 404-409).

“Future works may implement the proposed methodology in regions like the Aburrá Valley to compare the sustainable development priorities and establish the factors generating variations in the assignment of preferences. Likewise, future studies should consider the uncertainty and vagueness in expert judgment through linguistic evaluations, gray-based methods, and fuzzy logic systems.”

Finally, we thank the reviewers for their suggestions and observations, which helped to improve the quality of the article, both in its form and content, ensuring that it is an article of high impact for the academic and scientific community.

Reviewer 2 Report

Dear authors, 

I am glad I have this opportunity to review your manuscript. Based on its title/abstract it looks very interesting. Anyway, my comments/notes are the follows: 

  • Introduction
    • Stages of "assessing sustainable development using aggregated indices requires"... could be more described (specify what is the content of each stage by 1-2 sentences),
    • three dimensions could be illustrated by any figure, 
    • "For indicator weighting, several techniques..." - it is a poorly prepared overview of the methods used for criteria weighting (there are many other methods which could be also used). From my point of view, firstly should be these methods presented in general and after that, it could be more focused on one selected method (AHP), 
  • Indicator weighting for sustainable development
    • this is/could be a part of the methodology, not the theoretical section, 
    • Figure 2 - correct "Consistency accepted" where "y" is on a new line,
    • Table 1 - I expect more information about each criterion used (some base characteristics could be added, e.g. type of variable, source, some moment characteristics, ...), 
  • Pairwise Comparisons
    • I recommend changing the title of this section and do not divide next into short sub-sections (Normalized Matrix and Indicator Weighing Calculation, Consistency Test, ...). Is totally useless to have a section including one paragraph (Indicator Aggregation for SDG Weighing). They could be presented as steps of the calculation procedure, but for sure not as sub-sections.
    • you present your option (AHP, consistency test) as the only one possible option how to do it, there are many other approaches that are not considered or mentioned at least,
    • this section should be marked as "Methodology" or "Material and Methods", clearly should be divided into sections focusing on the subject (region), data and criteria description, methods used.
  • Results and Discussion
    • Table 4 - without any description,
    • Table 5 - totally useless (it is only an intermediate step that is ok, but it is not necessary to describe it by a table),
    • you are not interpreting the results obtained at all, you just put some numbers there and advocate the use of the AHP method (which should be a part of the previous section), 
    • "This work was focused on indicator..." - the same problem, this should be a part of other section, real discussion is missing (you do not have any results interpretation to discuss),
  • Conclusions
    • "However, the novelty of this study is providing an indicator weighting model that allows the assessment of sustainable development at the subnational level." - really? You just choose one method and applied it to one case study, this is novelty? 
    • "Future research should focus..." - it should be a part of this manuscript, in the current version it looks like you just select a part of your research for publishing it separately.

There are many methodological shortcomings that should be solved. The results should be better prepared and much more interpreted. Check also the language level.

Author Response

Dear Evay Yuan                                                                                              Medellín, October 5th, 2021

Economics
MDPI

We communicate with you through this letter in order to outline every change made in the article economies-1394433 “Application of AHP for the Weighting of Sustainable Development Indicators at the Subnational Level” based on the observations and suggestions provided by the reviewers (Reviewer 1, Reviewer 2). All changes and modifications made to the manuscript are highlighted in yellow in the revised version.

Comments from author to reviewers:
-Reviewer 1

The authors of this interesting article present a method for weighting indicators, more precisely for the construction of composite indices useful for assessing sustainable development at the subnational level, the case study discussed was Aburrá Valley region belonging to Antioquia state (Colombia). The topic of this manuscript falls within the objectives of the journal. The topic is well presented, and the English style and grammar are fine, as the text is easy to follow. However, further efforts are needed to make better use of the work. The authors are recommended as follows.

line 130: I would propose to report the three indicators that are not present in table 1.

R// A paragraph is added to explain why indicators related to SDG 9, SDG 13 and SDG 14 were not considered (lines 220-230):

“Note that indicators for SDG 9, SDG 13, and SDG 14 are not shown in Table 1 due to diverse causes. First, SDG 9 (Industry, innovation, and infrastructure) is not available in the MIO, and the availability of the indicator is essential for the construction of indices (Tanguay et al. 2010; Shaaban and Scheffran 2017). Second, indicators related to SDG 13 (Climate Action) are not available for all the municipalities of the Aburrá Valley, which is essential to compare them (Londoño 2018) and obtain a unified value for the region. Likewise, in the Aburrá Valley, no indicator covers the topic to be evaluated (Hák, Janousková, and Moldan 2016). Finally, SDG 14 (Life below water) does not apply to the Aburrá Valley because this region is located within the department of Antioquia, so it does not have beaches or access to the sea (Londoño and Cruz 2019).”

line 144 and 145: better clarify how the indicators are compared.

R// A column is added to Table 2 explaining how these indicators contribute directly to each SDG (line 246).

Topic preferences from CPS

Indicators

Indicator relationship with SDGs

SDGs to which the indicators belong

1. Health

X4

This indicator expresses the death of women during pregnancy, childbirth or during the next 42 days after the end of the pregnancy and is directly related to good health and well-being.

3. Good health and well-being

X5

It represents the number of deaths of children under 5 years and is directly related to good health and well-being.

X6

It represents the number of deaths among people with HIV-AIDS and is directly related to good health and well-being.

X23

It expresses the PM2.5, being the most important in urban pollution since they can penetrate the lungs and pose significant potential risks to health. This indicator is directly related to good health and well-being and sustainable cities and communities.

11. Sustainable cities and communities

2. Employment

X17

It expresses the percentage of workers who are part of the labor force and actively seeking work but are currently without it. It is directly related to decent work and economic growth.

8. Decent work and economic growth

X18

It expresses the labor force who are highly skilled but working in low-paying or low-skill jobs and part-time workers who would prefer to be full-time. It is directly related to decent work and economic growth.

X19

It represents a demographic index and expresses the proportion of people of non-working age, compared with the number of those of working age. It is directly related to decent work and economic growth.

3. Education

X7

It expresses the percentage of children between 7 and 11 years old who attend primary school. It is directly related to quality education.

4. Quality education

X8

It expresses the Percentage of children between 12 and 17 years old who attend secondary school. It is directly related to quality education.

X9

It measures the participation of youth and adults in higher education training programs. It is directly related to quality education.

X10

It expresses the relative magnitude of the illiterate population. It is directly related to quality education.

X11

It expresses the relative magnitude of the illiterate population. It is directly related to quality education.

4. Housing

X1

It determines whether the essential needs of the population are covered. It is directly related to poverty.

1. No poverty

X2

It is based on the indicators to determine a condition of misery. It is directly related to poverty.

X13

It represents the percentage of households with aqueduct coverage. It is directly related to clean water and sanitation.

6 Clean water and sanitation

X14

It represents the percentage of households with sewerage coverage. It is directly related to clean water and sanitation.

X15

It represents the percentage of homes with electricity. It is directly related to affordable and clean energy.

7. Affordable and clean energy

X16

It represents the percentage of homes with a gas connection. It is directly related to affordable and clean energy.

X28

It represents the percentage of households with internet connection. One of the targets of SDG 17 is associated with the issue of access to technology and networks.

17. Partnerships for the goals

5. Security

X26

It measures the intentional homicides, so it is directly related to peace, justice, and strong institutions.

16. Peace, justice, and strong institutions

X27

It measures the kidnappings, so it is directly related to peace, justice, and strong institutions.

6. Feeding

X3

It reveals the prevalence of food insecurity at moderate and severe levels. It is directly related to zero hunger.

2. Zero Hunger

7. Others: gender equality, environment, incomes

X19

It measures the percentage of women among the total members in the Municipal Councils and it is directly related to gender equality.

5. Gender equality

X20

It expresses the level of indebtedness of the municipalities, which limits the resources to reduce social inequalities and perform social investment. It is directly related to reduced inequalities.

10. Reduced inequalities

X21

It measures the percentage of households with internet coverage to reduce technological and social gaps. It is directly related to reduced inequalities.

X22

It expresses the progress of the urbanization of a territory, considering that an increasing value of the indicator threatens the sustainability of cities. It is directly related to sustainable cities and communities.

11. Sustainable cities and communities

X24

It measures the proper disposal of solid waste to avoid contamination and risks to human health and the environment. It is directly related to sustainable responsible consumption and production.

12. Responsible consumption and production

X25

It expresses the percentage of land in the municipality that is protected to avoid its depletion and use in highly polluting activities. It is directly related to life on land.

15. Life on land

line 243 to 251 : I would move the content of this part to an earlier section, even in the introduction.

R// It was decided to remove this paragraph from the results section. In the introduction, several paragraphs were added to deepen on the methods for weighting criteria (lines 84-115).

“For indicator weighting, weighting methods are classified into equal weighting and rank-order weighting (Shao et al. 2020; Si and Marjanovic-Halburd 2018). In equal weighting, indicator weights are equally assigned, which does not require stakeholder preferences; however, it ignores the relative importance of the criteria (Shao et al. 2020; Wang et al. 2009). In rank-order weighting methods, indicators weights are distributed and influenced by stakeholder perspectives (Si and Marjanovic-Halburd 2018). The rank-order weighting methods include subjective weighting method, objective weighting method and combination weighting method (Wang et al. 2009). Subjective weighting methods depend only on the preference of decision-makers, while objective weighting methods emphasize the statistical evaluation of the given data, complex calculation process and intensive data requirement (Shao et al. 2020). Combination weighting method is used to balance merits and limitations of objective and subjective weighting methods, but the process is complex and not widely-used (Si and Marjanovic-Halburd 2018).”

“Subjective weighting methods include Analytic Hierarchy Process (AHP), Analytic Network Process (ANP), Best Worst Method (BWM), Delphi method, DEMATEL, Direct given method, Eigenvector method, Expert judgement, Full consistency method Step-Wise Weight Assessment Ratio Analysis (SWARA), Fuzzy preference programming, Least-square method, Quality function deployment (QFD), Simos method, Simple Multi-Attribute Rating Technique (SMART), Swing method. Objective weighting methods include Data envelopment analysis (DEA), Divergence measure method, Entropy method, Hesitant fuzzy entropy measure, Least mean square (LMS) method, Linear programming technique for multidimensional analysis of preference (LINMAP), Maximizing deviation method, Minmax deviation method, Multiple correlation coefficient, Ordered weight, Preference selection index, Principal component analysis (PCA), Relative preference relation, Time sequence weight, Technique for Order of Preference for Similarity Ideal Solution TOPSIS method, Variation coefficient, Vertical and horizontal method. Combination weighting methods include Additive synthesis, AHP and entropy method, Direct given and ordered weight, Entropy method and divergence measure method, Expert judgement and entropy method, Expert judgement and maximizing deviation method, Expert judgement and maximizing distance method, Expert judgement and statistical variance method, Multiplication synthesis, Optimal weighting based on minimum bias, Optimal weighting based on sum of squares (Wang et al. 2009; L. J. Zhang et al. 2020).”

Comments from author to reviewers:
-Reviewer 2

I am glad I have this opportunity to review your manuscript. Based on its title/abstract it looks very interesting. Anyway, my comments/notes are the follows:

Introduction. Stages of "assessing sustainable development using aggregated indices requires"... could be more described (specify what is the content of each stage by 1-2 sentences).

R// The following paragraph is added to describe each of the steps required for assessing sustainable development (lines 60-83)

“The conceptual framework development establishes the approach to analyze the sustainable development assessment, establishing the backgrounds, contexts, and relationships between the economic, social, environmental, and institutional dimensions. The indicator selection for sustainable development assessment should provide an estimate of the achievement of the SDGs, quantify both the determinants and the final impacts of sustainability, and influence the decision-making process with the information they provide. The indicator selection can be supported by several methods and should adopt the following principles: systematic, consistency, independency, measurability, comparability  (Wang et al. 2009; Rigo et al. 2020; Shao et al. 2020). Imputation of missing data recognizes the existence of variables that cannot be observed or measured, so an imperfect or approximate measurement can be achieved based on the valid values of other variables or based on a sample. It tends to be recurrent in sustainable development assessment at the subnational level due to the lack of structured and formal sources of information related to the SDGs. Data standardization normalizes the data obtained from the selected indicators so they can be used for quantitative calculations (Shao et al. 2020). Indicator weighting allows a trade-off between multiple indicators and a balance between different sustainable development perspectives, so it is necessary to select the weighting method to define or quantify the importance/relevance of the selected indicators (Rigo et al. 2020; Németh et al. 2019). The aggregation implies aggregating the normalized information of the selected indicators considering their weighting for calculating a total sustainable development score. Sensitivity analysis represents a validation method mainly performed by varying the indicator weights and indicators values to establish how stable the results are when changing the conditions considered in the sustainable development assessment (Shao et al. 2020; Rigo et al. 2020; Allen et al. 2020).”

Introduction. • three dimensions could be illustrated by any figure

R// Figure 1 is added to illustrate the three dimensions and four dimensions of sustainable development (line 46)

(a)

(b)

Figure 1. Three dimensions of sustainable development (a); four dimensions of sustainable development (b)

Introduction. • "For indicator weighting, several techniques..." - it is a poorly prepared overview of the methods used for criteria weighting (there are many other methods which could be also used). From my point of view, firstly should be these methods presented in general and after that, it could be more focused on one selected method (AHP),

R// We added two paragraphs dedicated to an overview of the methods used for criteria weighting (lines 84-115)

“For indicator weighting, weighting methods are classified into equal weighting and rank-order weighting (Shao et al. 2020; Si and Marjanovic-Halburd 2018). In equal weighting, indicator weights are equally assigned, which does not require stakeholder preferences; however, it ignores the relative importance of the criteria (Shao et al. 2020; Wang et al. 2009). In rank-order weighting methods, indicators weights are distributed and influenced by stakeholder perspectives (Si and Marjanovic-Halburd 2018). The rank-order weighting methods include subjective weighting method, objective weighting method and combination weighting method (Wang et al. 2009). Subjective weighting methods depend only on the preference of decision-makers, while objective weighting methods emphasize the statistical evaluation of the given data, complex calculation process and intensive data requirement (Shao et al. 2020). Combination weighting method is used to balance merits and limitations of objective and subjective weighting methods, but the process is complex and not widely-used (Si and Marjanovic-Halburd 2018).”

“Subjective weighting methods include Analytic Hierarchy Process (AHP), Analytic Network Process (ANP), Best Worst Method (BWM), Delphi method, DEMATEL, Direct given method, Eigenvector method, Expert judgement, Full consistency method - Step-Wise Weight Assessment Ratio Analysis (SWARA), Fuzzy preference programming, Least-square method, Quality function deployment (QFD), Simos method, Simple Multi-Attribute Rating Technique (SMART), Swing method. Objective weighting methods include Data envelopment analysis (DEA), Divergence measure method, Entropy method, Hesitant fuzzy entropy measure, Least mean square (LMS) method, Linear programming technique for multidimensional analysis of preference (LINMAP), Maximizing deviation method, Minmax deviation method, Multiple correlation coefficient, Ordered weight, Preference selection index, Principal component analysis (PCA), Relative preference relation, Time sequence weight, Technique for Order of Preference for Similarity Ideal Solution TOPSIS method, Variation coefficient, Vertical and horizontal method. Combination weighting methods include Additive synthesis, AHP and entropy method, Direct given and ordered weight, Entropy method and divergence measure method, Expert judgement and entropy method, Expert judgement and maximizing deviation method, Expert judgement and maximizing distance method, Expert judgement and statistical variance method, Multiplication synthesis, Optimal weighting based on minimum bias, Optimal weighting based on sum of squares (Wang et al., 2009; Zhang et al., 2020).”

Likewise, a paragraph is added to describe the functionality and practicality of the AHP for the weighting of indicators (lines 141-147):

“Moreover, the AHP is simple to understand and apply to complex issues using uses a 9-point scale to compare criteria relative importance. The AHP decomposes a large problem into smaller sub-problems at hierarchical levels, introduces the comparative importance of the criteria, showing a more reliable representation of the decision goal. The AHP is applicable for quantitative and qualitative criteria and checks the consistency of the decision, thus reducing the bias in the decision-making progression (Shao et al., 2020; Si & Marjanovic-Halburd, 2018).”

Indicator weighting for sustainable development. •         this is/could be a part of the methodology, not the theoretical section,

R // The name of the section “Indicator weighting for sustainable development” is changed to “Methodology” (line 180)

Indicator weighting for sustainable development. •         Figure 2 - correct "Consistency accepted" where "y" is on a new line,

R // The word “Consistency” is corrected to present it in a single line (line 202)

Indicator weighting for sustainable development. •         Table 1 - I expect more information about each criterion used (some base characteristics could be added, e.g. type of variable, source, some moment characteristics, ...),

R // A column with the description of each indicator is added to Table 1 (line 231)

Sustainable Development Goals

Goal Code

Indicator

Indicator description

Indicator Code

1. No poverty

SDG1

Percentage of households with at least one unsatisfied basic need (Poverty)

It determines whether the essential needs of the population are covered. It is based on the indicators of inadequate housing, housing with critical overcrowding, housing with deficient services, housing with high economic dependence, housing with school-age children who do not attend school.

X1

Percentage of households with two or more unsatisfied basic needs (Misery)

It is based on the indicators of inadequate housing, housing with critical overcrowding, housing with deficient services, housing with high economic dependence, housing with school-age children who do not attend school. In case of not fulfilling two or more of these, it would be in a condition of misery.

X2

2. Zero Hunger

SDG2

Percentage of households with food insecurity

It reveals the prevalence of food insecurity at moderate and severe levels. The information is collected through surveys.

X3

3. Good health and well-being

SDG3

Maternal mortality rate

It measures the death of a woman during pregnancy, childbirth or during the next 42 days after the end of the pregnancy. It is calculated based on every 100.000 live births.

X4

Under-five mortality rate

It represents the number of deaths of children under 5 years for every 1.000 live births for a given year, in each country, territory, or geographic area.

X5

Mortality rate for HIV-AIDS

It represents the number of deaths among people with HIV-AIDS per 100.000 inhabitants for a given year, in each country, territory, or geographic area.

X6

4. Quality education

SDG4

School attendance rate in primary school

Percentage of children between 7 and 11 years old who attend primary school

X7

Secondary school attendance rate

Percentage of children between 12 and 17 years old who attend secondary school

X8

Schooling rate in higher education

It shows the relationship between students enrolled at the undergraduate level (technical professional, technological, and university) and the projected population between 17 and 21 years old. Therefore, it measures the participation of youth and adults in higher education training programs.

X9

Illiteracy rate from 10 to 14 years old

It expresses the relative magnitude of the illiterate population and calculates the population between 10 and 14 years old who cannot read and write divided by the population greater than or equal to 15 years.

X10

Illiteracy rate in people older than 15 years

It expresses the relative magnitude of the illiterate population and calculates the population greater than or equal to 15 years old who cannot read and write divided by the population greater than or equal to 15 years.

X11

5. Gender equality

SDG5

Percentage of women in municipal councils

Percentage of women among the total members in the Municipal Councils.

X12

6 Clean water and sanitation

SDG6

Aqueduct coverage

Percentage of households with aqueduct coverage

X13

Sewer coverage

Percentage of households with sewerage coverage

X14

7. Affordable and clean energy

SDG7

Energy coverage

Percentage of homes with electricity

X15

Gas connection coverage

Percentage of homes with a gas connection

X16

8. Decent work and economic growth

SDG8

Unemployment rate

Percentage of workers who are part of the labor force and actively seeking work but are currently without it

X17

Underemployment rate

Looks at how well the labor force is being used in terms of skills, experience, and availability to work. People who are classified as underemployed include workers who are highly skilled but working in low-paying or low-skill jobs and part-time workers who would prefer to be full-time.

X18

Dependency ratio

Represents a demographic index and expresses the proportion of people of non-working age, compared with the number of those of working age.

X19

10. Reduced inequalities

SDG10

Indebtedness index

It expresses the level of indebtedness of the municipalities. A growing value of this indicator limits the resources available to reduce social inequalities and perform social investment.

X20

Internet coverage

Percentage of households with internet coverage, which is necessary to reduce technological and social gaps.

X21

11. Sustainable cities and communities

SDG11

Percentage of rural land

It expresses the progress of the urbanization of a territory, considering that an increasing value of the indicator threatens the sustainability of cities.

X22

Concentration of Particulate Material PM2,5

It expresses the material with a particle size of fewer than 2.5 microns, known as PM2.5, being the most important in urban pollution since they can penetrate the lungs and pose significant potential risks to health.

X23

12. Responsible consumption and production

SDG12

Percentage of solid waste used

It expresses the percentage of solid waste that companies separate and dispose of in places specially designed to avoid contamination and risks to human health and the environment.

X24

15. Life on land

SDG15

Percentage of soil protection

It expresses the percentage of land in the municipality that is protected to avoid its depletion and use in highly polluting activities.

X25

16. Peace, justice and strong institutions

SDG16

Homicide rate

It expresses the intentional homicides per 100.000 inhabitants in a territory in a period.

X26

Kidnapping rate

It expresses the kidnappings per 100.000 inhabitants in a territory in a period.

X27

17. Partnerships for the goals

SDG17

Internet penetration rate

Percentage of households with at least one member between the ages of 16 and 74 who have internet access or percentage of households with broadband connection.

X28

Pairwise Comparisons •              I recommend changing the title of this section and do not divide next into short sub-sections (Normalized Matrix and Indicator Weighing Calculation, Consistency Test, ...). Is totally useless to have a section including one paragraph (Indicator Aggregation for SDG Weighing). They could be presented as steps of the calculation procedure, but for sure not as sub-sections.

R // The structure of the Methodology section is modified considering only two subsections: 2.1. Data and Indicator Description (line 204) and 2.2 AHP for indicator weighting (line 258)

Pairwise Comparisons •              you present your option (AHP, consistency test) as the only one possible option how to do it, there are many other approaches that are not considered or mentioned at least,

R// We added two paragraphs dedicated to an overview of the methods used for criteria weighting (lines 84-115)

“For indicator weighting, weighting methods are classified into equal weighting and rank-order weighting (Shao et al. 2020; Si and Marjanovic-Halburd 2018). In equal weighting, indicator weights are equally assigned, which does not require stakeholder preferences; however, it ignores the relative importance of the criteria (Shao et al. 2020; Wang et al. 2009). In rank-order weighting methods, indicators weights are distributed and influenced by stakeholder perspectives (Si and Marjanovic-Halburd 2018). The rank-order weighting methods include subjective weighting method, objective weighting method and combination weighting method (Wang et al. 2009). Subjective weighting methods depend only on the preference of decision-makers, while objective weighting methods emphasize the statistical evaluation of the given data, complex calculation process and intensive data requirement (Shao et al. 2020). Combination weighting method is used to balance merits and limitations of objective and subjective weighting methods, but the process is complex and not widely-used (Si and Marjanovic-Halburd 2018).”

“Subjective weighting methods include Analytic Hierarchy Process (AHP), Analytic Network Process (ANP), Best Worst Method (BWM), Delphi method, DEMATEL, Direct given method, Eigenvector method, Expert judgement, Full consistency method - Step-Wise Weight Assessment Ratio Analysis (SWARA), Fuzzy preference programming, Least-square method, Quality function deployment (QFD), Simos method, Simple Multi-Attribute Rating Technique (SMART), Swing method. Objective weighting methods include Data envelopment analysis (DEA), Divergence measure method, Entropy method, Hesitant fuzzy entropy measure, Least mean square (LMS) method, Linear programming technique for multidimensional analysis of preference (LINMAP), Maximizing deviation method, Minmax deviation method, Multiple correlation coefficient, Ordered weight, Preference selection index, Principal component analysis (PCA), Relative preference relation, Time sequence weight, Technique for Order of Preference for Similarity Ideal Solution TOPSIS method, Variation coefficient, Vertical and horizontal method. Combination weighting methods include Additive synthesis, AHP and entropy method, Direct given and ordered weight, Entropy method and divergence measure method, Expert judgement and entropy method, Expert judgement and maximizing deviation method, Expert judgement and maximizing distance method, Expert judgement and statistical variance method, Multiplication synthesis, Optimal weighting based on minimum bias, Optimal weighting based on sum of squares (Wang et al., 2009; Zhang et al., 2020).”

Likewise, a paragraph is added to describe the functionality and practicality of the AHP for the weighting of indicators (lines 141-147):

“Moreover, the AHP is simple to understand and apply to complex issues using uses a 9-point scale to compare criteria relative importance. The AHP decomposes a large problem into smaller sub-problems at hierarchical levels, introduces the comparative importance of the criteria, showing a more reliable representation of the decision goal. The AHP is applicable for quantitative and qualitative criteria and checks the consistency of the decision, thus reducing the bias in the decision-making progression (Shao et al., 2020; Si & Marjanovic-Halburd, 2018).”

Indicator weighting for sustainable development. •         this section should be marked as "Methodology" or "Material and Methods", clearly should be divided into sections focusing on the subject (region), data and criteria description, methods used.

R// The name of the section “Indicator weighting for sustainable development” is changed to “Methodology” (line 180). The structure of the Methodology section is modified considering only two subsections: 2.1. Data and Indicator Description (line 204) and 2.2 AHP for indicator weighting (line 258)

Results and Discussion •             Table 4 - without any description,

R // We improve and deepen the description of Table 4 (lines 314-331):

“As a result of the AHP method, Table 4 shows the matrix (A) of paired comparisons, which contains values from a 9-point scale to compare indicators' relative importance that corresponds to the original scale of preferences of Saatty.”

“These values reflect the importance of indicators defined by expert criteria by comparing elements of a row with elements of a column. For example, when comparing X4 with X20 (value of row 4 and column 20), a value of 9 was assigned, which means that the indicator X4 (Maternal mortality rate) is of absolute importance concerning the indicator X19 (Indebtedness index), then absolutely favoring the good health and well-being of pregnant women concerning the level of indebtedness of the municipalities, that is, prioritizing the social dimension over the economic and institutional dimension. Respectively, the value of the element in row 20 and column 4 corresponds to the inverse value of 9 (0,11).”

“Likewise, when comparing X4 with X23 (row 4 and column 23), a value of 1 was assigned, which means that the indicator X4 (Maternal mortality rate) is equally important as the indicator X23 (Concentration of Particulate Material PM2.5) since air quality directly affects the health of citizens, specifically those who are at risk of respiratory diseases. Therefore, it is considered that both indicators contribute equally to the social dimension. Respectively, the element in row 23 and column 4 corresponds to the inverse of 1, which is also equal to 1.”

Results and Discussion •             Table 5 - totally useless (it is only an intermediate step that is ok, but it is not necessary to describe it by a table),

R// Table 5 is eliminated, and it is mentioned as a complement to the description of Table 4 how the normalized values are calculated (lines 334-336).

“Equation (1) is applied to the values of the matrix (A) of paired comparisons to obtain the Normalized Matrix, and the Indicator Weighing Calculation is obtained based on that matrix using Equation (2).”

Results and Discussion • you are not interpreting the results obtained at all, you just put some numbers there and advocate the use of the AHP method (which should be a part of the previous section),

R// Three paragraphs are added to interpret the results obtained (lines 355-378):

“The results of this work show that the sustainable development priorities for the municipalities of the Aburrá Valley should focus primarily on the health issue, defined by the CPS, since the indicators with the most considerable weight are related to SDG3, as they are the maternal mortality rate, mortality rate for HIV-AIDS, and the under-five mortality rate. The health issue also includes an indicator of SDG 11 related to concentrations of particulate matter PM2.5 since poor air quality constitutes a public health problem in the Aburrá Valley. This situation is relevant because this region is surrounded by mountains that do not allow adequate air circulation, concentrating particulate material inside the valley.”

“Moreover, this study prioritizes the issue of employment represented in the indicators that contribute to SDG8. In this sense, the unemployment rate, underemployment rate, and dependency rate worsened due to the confinement generated by the Covid-19 pandemic; therefore, it will continue as a priority in the next quality of life surveys in this region. Similarly, the issue of education occupies the third place of preferences with the indicators related to SDG4, such as the coverage of primary, secondary and higher education, and illiteracy rate indicators. The prioritization of this indicator indicates that governments of the municipalities of the Aburrá Valley must provide proper conditions for citizens to access education, especially higher education, to train citizens for the labor market that continually requires a more skilled workforce.”

“Accordingly, the indicators with the most considerable weight reflect the priorities of sustainable development in the Aburrá Valley corresponding to SDG3 (Health and well-being), SDG8 (Decent work and growth economic), and SDG4 (Quality education). Therefore, development plans, programs, and the municipal budget of these municipalities should focus on improving health, education, and employment issues.”

Results and Discussion •             "This work was focused on indicator..." - the same problem, this should be a part of other section, real discussion is missing (you do not have any results interpretation to discuss),

R// The entire paragraph was moved and placed at the end of the Introduction section (lines 156-173)

Conclusions •   "However, the novelty of this study is providing an indicator weighting model that allows the assessment of sustainable development at the subnational level." - really? You just choose one method and applied it to one case study, this is novelty?

R// The novelty of this study is highlighted in the conclusions in a new paragraph added (lines 396-401):

“This study is novel because it allows prioritizing sustainable development dimensions through a methodology that articulates the technical approach derived from the 17 SDGs of the 2030 agenda (objective) with a participatory approach represented by the CPS (subjective) where citizens prioritize the most relevant life-quality issues. Moreover, a multi-criteria method based on the AHP supports the proposed methodology to lead political decision-making.”

Conclusions •   "Future research should focus..." - it should be a part of this manuscript, in the current version it looks like you just select a part of your research for publishing it separately.

R// The future research suggestion is modified in the conclusions, showing future lines of research that do not depend directly on the stages of our research (lines 404-409).

“Future works may implement the proposed methodology in regions like the Aburrá Valley to compare the sustainable development priorities and establish the factors generating variations in the assignment of preferences. Likewise, future studies should consider the uncertainty and vagueness in expert judgment through linguistic evaluations, gray-based methods, and fuzzy logic systems.”

Finally, we thank the reviewers for their suggestions and observations, which helped to improve the quality of the article, both in its form and content, ensuring that it is an article of high impact for the academic and scientific community.

Round 2

Reviewer 2 Report

Dear authors, 

thanks for the time you spent incorporating my comments, I really appreciate it. I consider each of them discussed and/or incorporated. Anyway, I recommend some other "light" changes":

  • introduction - line 97 - we should recognize between methods of weight determination (SWARA, ...) and multi-criteria methods (TOPSIS, ...), please rewritten this section (see a comparison of some these methods to check the differences here:  https://www.worldscientific.com/doi/10.1142/S021962201950041X
  • introduction - it is too long, so it would be useful to create a separate section "Theoretical background", the last paragraph of the introduction would be the same ("Therefore, this paper... see line 174),
  • table 1/2 - the current version is too big, so I recommend two possibilities: a) replace it to the annex, b) create a shorter version in the text + add information that more comprehensive information could be found in the annex (it should be also replaced) 

Author Response

Dear Evay Yuan                                                                                             Economics
MDPI

We communicate with you through this letter in order to outline every change made in the article economies-1394433 “Application of AHP for the Weighting of Sustainable Development Indicators at the Subnational Level” based on the observations and suggestions provided by the Reviewer 2. All changes and modifications made to the manuscript are highlighted in blue in the revised version.

Comments from author to reviewers:
-Reviewer 2

Introduction - line 97 - we should recognize between methods of weight determination (SWARA, ...) and multi-criteria methods (TOPSIS, ...), please rewritten this section (see a comparison of some these methods to check the differences here: https://www.worldscientific.com/doi/10.1142/S021962201950041X.

R// The article suggested by the reviewer was analyzed and the paragraph was modified to state the methods of weight determination (lines 126-134)

“Subjective weighting methods include Analytic Hierarchy Process (AHP), Delphi method, Pairwise comparison, Ranking method, SIMOS method, Simple Multi-Attribute Rating Technique (SMART), Step-Wise Weight Assessment Ratio Analysis (SWARA), Swing Weighting (SW) (Vavrek 2019; Wang et al. 2009). Objective weighting methods include Criteria Importance Through Intercriteria Correlation (CRITIC), Entropy method, Mean Weight (MW), Standard Deviation (SD), Coefficient of Variance (CV), Statistical Variance Procedure (SVP), Integrated Determination of Objective Criteria Weights (IDOCRIW), Criterion Impact Loss (CILOS), Principal component analysis (PCA) (Vavrek 2019; Kumar et al. 2009). “

In this way, the works of (Vavrek 2019), (Wang et al. 2009) and (Kumar et al. 2009) were considered in the paper.

  • Kumar, Rajesh, H R Murty, S K Gupta, and A K Dikshit. 2009. “An Overview of Sustainability Assessment Methodologies.” Ecological Indicators 9: 189–212. https://doi.org/10.1016/j.ecolind.2008.05.011.

  • Vavrek, Roman. 2019. “Evaluation of the Impact of Selected Weighting Methods on the Results of the TOPSIS Technique.” International Journal of Information Technology & Decision Making 18 (6): 1821–43. https://doi.org/10.1142/S021962201950041X.

  • Wang, Jiang Jiang, You Yin Jing, Chun Fa Zhang, and Jun Hong Zhao. 2009. “Review on Multi-Criteria Decision Analysis Aid in Sustainable Energy Decision-Making.” Renewable and Sustainable Energy Reviews 13 (9): 2263–78. https://doi.org/10.1016/j.rser.2009.06.021.

Introduction - it is too long, so it would be useful to create a separate section "Theoretical background", the last paragraph of the introduction would be the same ("Therefore, this paper... see line 174),

R// We create a separate section “2. Theoretical Background" in which the stages for sustainable development assessment (line 83) are addressed, then providing a shorter Introduction.

Table 1/2 - the current version is too big, so I recommend two possibilities: a) replace it to the annex, b) create a shorter version in the text + add information that more comprehensive information could be found in the annex (it should be also replaced) 

R // Appendix A (line 407) is created to show there in Table A1 and Table A2 the detailed information presented in Table 1 and Table 2.

Appendix A

Table A1. Description of indicators for assessing sustainable development.

Indicator Code

Indicator description

X1

It determines whether the essential needs of the population are covered. It is based on the indicators of inadequate housing, housing with critical overcrowding, housing with deficient services, housing with high economic dependence, housing with school-age children who do not attend school.

X2

It is based on the indicators of inadequate housing, housing with critical overcrowding, housing with deficient services, housing with high economic dependence, housing with school-age children who do not attend school. In case of not fulfilling two or more of these, it would be in a condition of misery.

X3

It reveals the prevalence of food insecurity at moderate and severe levels. The information is collected through surveys.

X4

It measures the death of a woman during pregnancy, childbirth or during the next 42 days after the end of the pregnancy. It is calculated based on every 100.000 live births.

X5

It represents the number of deaths of children under 5 years for every 1.000 live births for a given year, in each country, territory, or geographic area.

X6

It represents the number of deaths among people with HIV-AIDS per 100.000 inhabitants for a given year, in each country, territory, or geographic area.

X7

Percentage of children between 7 and 11 years old who attend primary school

X8

Percentage of children between 12 and 17 years old who attend secondary school

X9

It shows the relationship between students enrolled at the undergraduate level (technical professional, technological, and university) and the projected population between 17 and 21 years old. Therefore, it measures the participation of youth and adults in higher education training programs.

X10

It expresses the relative magnitude of the illiterate population and calculates the population between 10 and 14 years old who cannot read and write divided by the population greater than or equal to 15 years.

X11

It expresses the relative magnitude of the illiterate population and calculates the population greater than or equal to 15 years old who cannot read and write divided by the population greater than or equal to 15 years.

X12

Percentage of women among the total members in the Municipal Councils.

X13

Percentage of households with aqueduct coverage

X14

Percentage of households with sewerage coverage

X15

Percentage of homes with electricity

X16

Percentage of homes with a gas connection

X17

Percentage of workers who are part of the labor force and actively seeking work but are currently without it

X18

Looks at how well the labor force is being used in terms of skills, experience, and availability to work. People who are classified as underemployed include workers who are highly skilled but working in low-paying or low-skill jobs and part-time workers who would prefer to be full-time.

X19

Represents a demographic index and expresses the proportion of people of non-working age, compared with the number of those of working age.

X20

It expresses the level of indebtedness of the municipalities. A growing value of this indicator limits the resources available to reduce social inequalities and perform social investment.

X21

Percentage of households with internet coverage, which is necessary to reduce technological and social gaps.

X22

It expresses the progress of the urbanization of a territory, considering that an increasing value of the indicator threatens the sustainability of cities.

X23

It expresses the material with a particle size of fewer than 2.5 microns, known as PM2.5, being the most important in urban pollution since they can penetrate the lungs and pose significant potential risks to health.

X24

It expresses the percentage of solid waste that companies separate and dispose of in places specially designed to avoid contamination and risks to human health and the environment.

X25

It expresses the percentage of land in the municipality that is protected to avoid its depletion and use in highly polluting activities.

X26

It expresses the intentional homicides per 100.000 inhabitants in a territory in a period.

X27

It expresses the kidnappings per 100.000 inhabitants in a territory in a period.

X28

Percentage of households with at least one member between the ages of 16 and 74 who have internet access or percentage of households with broadband connection.

Table A2. Relationship of indicators with SDGs

Indicators

Indicator relationship with SDGs

SDG

X4

This indicator expresses the death of women during pregnancy, childbirth or during the next 42 days after the end of the pregnancy and is directly related to good health and well-being.

SDG3

X5

It represents the number of deaths of children under 5 years and is directly related to good health and well-being.

X6

It represents the number of deaths among people with HIV-AIDS and is directly related to good health and well-being.

X23

It expresses the PM2.5, being the most important in urban pollution since they can penetrate the lungs and pose significant potential risks to health. This indicator is directly related to good health and well-being and sustainable cities and communities.

SDG11

X17

It expresses the percentage of workers who are part of the labor force and actively seeking work but are currently without it. It is directly related to decent work and economic growth.

SDG8

X18

It expresses the labor force who are highly skilled but working in low-paying or low-skill jobs and part-time workers who would prefer to be full-time. It is directly related to decent work and economic growth.

X19

It represents a demographic index and expresses the proportion of people of non-working age, compared with the number of those of working age. It is directly related to decent work and economic growth.

X7

It expresses the percentage of children between 7 and 11 years old who attend primary school. It is directly related to quality education.

SDG4

X8

It expresses the Percentage of children between 12 and 17 years old who attend secondary school. It is directly related to quality education.

X9

It measures the participation of youth and adults in higher education training programs. It is directly related to quality education.

X10

It expresses the relative magnitude of the illiterate population. It is directly related to quality education.

X11

It expresses the relative magnitude of the illiterate population. It is directly related to quality education.

X1

It determines whether the essential needs of the population are covered. It is directly related to poverty.

SDG1

X2

It is based on the indicators to determine a condition of misery. It is directly related to poverty.

X13

It represents the percentage of households with aqueduct coverage. It is directly related to clean water and sanitation.

SDG6

X14

It represents the percentage of households with sewerage coverage. It is directly related to clean water and sanitation.

X15

It represents the percentage of homes with electricity. It is directly related to affordable and clean energy.

SDG7

X16

It represents the percentage of homes with a gas connection. It is directly related to affordable and clean energy.

X28

It represents the percentage of households with internet connection. One of the targets of SDG 17 is associated with the issue of access to technology and networks.

SDG17

X26

It measures the intentional homicides, so it is directly related to peace, justice, and strong institutions.

SDG16

X27

It measures the kidnappings, so it is directly related to peace, justice, and strong institutions.

X3

It reveals the prevalence of food insecurity at moderate and severe levels. It is directly related to zero hunger.

SDG2

X19

It measures the percentage of women among the total members in the Municipal Councils and it is directly related to gender equality.

SDG5

X20

It expresses the level of indebtedness of the municipalities, which limits the resources to reduce social inequalities and perform social investment. It is directly related to reduced inequalities.

SDG10

X21

It measures the percentage of households with internet coverage to reduce technological and social gaps. It is directly related to reduced inequalities.

X22

It expresses the progress of the urbanization of a territory, considering that an increasing value of the indicator threatens the sustainability of cities. It is directly related to sustainable cities and communities.

SDG11

X24

It measures the proper disposal of solid waste to avoid contamination and risks to human health and the environment. It is directly related to sustainable responsible consumption and production.

SDG12

X25

It expresses the percentage of land in the municipality that is protected to avoid its depletion and use in highly polluting activities. It is directly related to life on land.

SDG15

Finally, we thank the reviewers for their suggestions and observations, which helped to improve the quality of the article, both in its form and content, ensuring that it is an article of high impact for the academic and scientific community.
